# PROSAFEPRUNE: PROJECTED SAFETY PRUNING FOR MITIGATING OVER-REFUSAL IN LLMS

Zijun Chen[1,2], Wenbo Hu[1,*], Ya Li[2], Lei Miao[2], Guoping Hu[2], and Richang Hong[1]

[1]Hefei University of Technology , `chenzijun@mail.hfut.edu.cn`,
`hongrc.hfut@gmail.com`
[2]iFLYTEK Research, Hefei, China ,{`yali, leimou, guopinghu`}`@iflytek.com`
[*]Corresponding author: `wenbohu@hfut.edu.cn`

## ABSTRACT

Large Language Models (LLMs) excel in various domains, but their safe deployment faces the challenge of balancing safety and utility. Existing alignment strategies often strengthen refusal mechanisms to reduce harmful outputs, but harmless instructions with superficial risky words are mistakenly rejected, which is known as *over-refusal*. This work first reveals that over-refusal stems from a *cognitive bias* in the model's internal representation space: LLMs naturally encode safety attributes in hidden states, and pseudo-harmful instructions overlap with harmful features, causing over-harmful encoding. To address this, we propose ProSafePrune, a subspace-projected low-rank parameter pruning framework for mitigating over-refusal. By projecting pseudo-harmful features into subspaces and removing low-rank directions corresponding to harmful components in the most discriminative layers, we significantly reduce over-refusal while preserving the model's ability to reject genuinely harmful requests, improving performance on general tasks. In experiments, across different models, our method significantly lowers the average false rejection rate while slightly improving general task performance. Code: `https://github.com/hfutml/PROSAFEPRUNE`.

## 1 INTRODUCTION

Large Language Models (LLMs) achieve strong performance across domains, yet their safe deployment hinges on balancing safety and usefulness (Bai et al., 2022; Bianchi et al., 2024; Dai et al., 2024; Huang et al., 2024; Zou et al., 2024; Wang et al., 2026; Gou et al., 2025). Current alignment practices typically strengthen refusal mechanisms to suppress harmful outputs, but this often results in over-refusal (Röttger et al., 2024; Zhang et al., 2025a)—rejecting not only genuinely harmful requests but also harmless ones that merely contain superficially risky cues. This undermines usability and has long been viewed as a side effect of overly conservative alignment.

Building on this observation, prior research has explored two main lines of mitigation. Training-based methods (Zhang et al., 2025b; Dabas et al., 2025; Wu et al., 2025) can be effective, but they demand additional data and computation, limiting practicality. Training-free methods (Cao et al., 2025; Shi et al., 2024) provide greater flexibility, yet they often fail to resolve the underlying bias and frequently introduce extra inference-time overhead. These limitations highlight the need for an approach that is both lightweight and able to directly address the root cause of over-refusal.

In contrast, we firstly reveal that over-refusal is not merely a failure of existing alignment pipelines, but rather a manifestation of an inherent *cognitive bias* in the model's internal representation space. Prior studies have shown that hidden states of LLMs naturally encode safety attributes of input instructions (Li et al., 2025; Zhao et al., 2025). However, pseudo-harmful instructions often project simultaneously onto both the harmful subspace and the harmless subspace. Under excessive safety fine-tuning, this natural overlap becomes distorted: the harmful projection is disproportionately amplified while the harmless one is suppressed. As a result, benign instructions are over-encoded with harmful features, shifting the internal decision boundary and ultimately leading to false rejections.

Interestingly, this phenomenon—which we term *over-harmful encoding*—also provides insight into the origin of the alignment tax (Huang et al., 2025), where models are pushed into an overly cautious regime that further constrains their general task performance.

In this work, we introduce a new perspective for mitigating over-refusal by directly acting on the parameter space of the model, rather than relying only on activation-level interventions or costly re-training. Our key insight is that over-refusal arises because model parameters contain excessive low-rank components that disproportionately encode harmfulness, which in turn causes pseudo-harmful instructions to be misclassified. Unlike training-based methods, our approach requires only a small amount of auxiliary data and avoids expensive fine-tuning; unlike existing training-free methods, it does not depend on repeated inference-time adjustments that introduce extra overhead while failing to address the underlying cognitive bias. Instead, we construct subspace projections to disentangle pseudo-harmful and harmful features, and apply a low-rank pruning strategy to selectively remove harmful components from the most discriminative layers. Since these directions occupy only a tiny fraction of the parameter space, pruning them alleviates over-refusal while minimally affecting the model's overall behavior, thus striking a better balance between safety and usefulness.

In summary, our contributions are as follows:

- We reveal that over-refusal in LLMs is caused by a cognitive bias in internal representations, due to harmless instructions being over-encoded with harmful features.

- To address this, we introduce ProSafePrune which prunes low-rank harmful components from the model's parameters, reducing over-refusal without additional inference-time cost.

- We identify that the model's internal over-harmful encoding is closely related to the alignment tax, which limits general performance. Our method effectively corrects this issue, leading to improved performance on general task.

## 2 RELATED WORK

### 2.1 OVER-REFUSAL IN LLMS

Safety alignment technologies for LLMs are critical to mitigating the risks of malicious instructions. Mainstream approaches include Supervised Fine-Tuning (SFT) and Reinforcement Learning from Human Feedback (RLHF), with the goal of preventing models from generating harmful content or responding to malicious requests (Bai et al., 2022; Bianchi et al., 2024). However, existing safety alignment solutions generally suffer from an over-defense issue: models unnecessarily reject pseudo-harmful prompts—inputs that are semantically harmless but contain superficially risky vocabulary. The academic community defines this phenomenon as over-refusal or exaggerated safety (Röttger et al., 2024; Varshney et al., 2024). For instance, the LLM incorrectly refuses benign requests like "how to kill the lights in the room" solely due to the word "kill" (Cao et al., 2025). Such behavior severely undermines model utility and highlights the intractable trade-off between harmlessness and helpfulness.

Fundamentally, over-refusal stems from the model's generalization bias toward safety signals: during the safety alignment process, models tend to forcibly associate risky vocabulary with refusal behavior, while ignoring contextual semantic differences. Early studies have confirmed that this bias is widespread across mainstream LLMs and exhibits a positive correlation with the model's jailbreak resistance—the more a model prioritizes defending against malicious instructions, the more likely it is to over-refuse benign prompts (Huang et al., 2024). This finding has driven systematic academic research on over-refusal, covering areas such as evaluation benchmark construction and mitigation method design. While these findings are often interpreted as a limitation of external alignment strategies, we argue that over-refusal reflects a deeper cognitive bias in the model's internal representation space. In other words, the problem is not just how alignment is applied at the output level, but how safety attributes are encoded and distorted within the hidden states themselves, leading to over-harmful encoding of pseudo-harmful instructions.

### 2.2 OVER-REFUSAL MITIGATION

Existing mitigation methods fall into two categories: training-based and training-free.

**Training-based.** These methods correct over-refusal by adjusting model parameters, recalibrating the safety decision boundary with pseudo-harmful data. Safety Patching (Zhao et al., 2024) generates gradient-based patches—enhancing refusal for harmful prompts and suppressing it for pseudo-harmful ones—and integrates them into specific layers. Zeng et al. show that supervised fine-tuning (SFT) on paired pseudo-harmful data reduces over-refusal. Dabas et al. (2025) further identify safety-critical layers, extract refusal direction vectors, and fine-tune only those layers via a dedicated refusal loss.

**Training-free.** These methods avoid parameter updates and intervene at inference. Ray & Bhalani (2024) propose a prompt-based strategy. Shi et al. (2024) (Self-CD) decode prompts twice—with and without a safety prompt—and suppress refusal tokens via probability comparison. Early work (Zou et al., 2023a) identified a refusal-related activation direction, which can be manipulated to alter model behavior. Building on this, Cao et al. (2025) extract refusal steering vectors from harmful vs. benign activations, locate safety-critical layers via vocabulary projection, and design a cosine-similarity classifier for adaptive adjustment. Wang et al. (2025) separate true vs. false refusal vectors by orthogonalization, ablating only the false ones to preserve genuine safety.

While effective, training-based methods require costly retraining, and training-free ones add inference overhead without addressing the root cause. In contrast, our pruning approach is data-efficient, cost-free at inference, and directly corrects the internal cognitive bias driving over-refusal.

## 3 METHODOLOGY

We tackle over-refusal by directly addressing the model's internal over-harmful encoding. First, we analyze layer-wise activations to reveal how pseudo-harmful instructions overlap with harmful features. Then, we apply subspace-based low-rank pruning to remove only the harmful components within pseudo-harmful directions, preserving genuine refusal ability while reducing over-refusal. Pruning is applied to layers with the strongest feature separability, ensuring minimal impact on overall utility.

### 3.1 OVER-HARMFULNESS ENCODING ACROSS MODEL LAYERS

Numerous studies have shown that the hidden states of LLMs contain rich information (Chen et al., 2026; Burns et al., 2023; Chen et al., 2025; Skean et al., 2025), including representations of harmfulness towards instructions. This encoding of harmfulness may influence the model's final output decisions, with excessive harmful encoding potentially being a contributing factor to the phenomenon of over-refusal. To investigate the underlying mechanism of this phenomenon, we analyze the activation distributions of LLMs at different layers in response to pseudo-harmful instructions, aiming to uncover the evolution of harmfulness representations within the model's internal layers.

To analyze the harmfulness encoding of instructions, we use probes (Alain & Bengio, 2016), which are trained to predict the harmfulness of instructions based on activation vectors extracted from the model's layers. Specifically, for each weight matrix $W_{l,m}$ at the $l$-th layer of the model, we extract the activation output for an input instruction $x$, which is then compressed into a vector via mean pooling. A logistic regression classifier is trained on safe and harmful instructions, and this classifier is used to predict the harmfulness of pseudo-harmful instructions. For more details, see Appendix A.1.

Figure 1 shows the layer-wise harmfulness encoding of pseudo-harmful instructions in LLaMA-2-7B and LLaMA-3-8B. We observe that LLaMA-2 exhibits stronger harmfulness signals in deeper layers, which correlates with its higher false rejection rate compared to LLaMA-3.

### 3.2 SUBSPACE-BASED PARAMETER PRUNING

Probing experiments reveal that in severely over-refusal models, pseudo-harmful instructions retain excessive harmful components in their high-level hidden representations, overlapping with genuinely harmful features. Despite the model's ability to leverage global knowledge to distinguish harmless from those instructions, deep parameter mappings continue to amplify harmful features. Over-refusal, therefore, results from over-harmfulized parameters rather than a feature-level artifact. While post-hoc interventions (e.g., activation editing, vector shifting (Cao et al., 2025; Wang et al.,

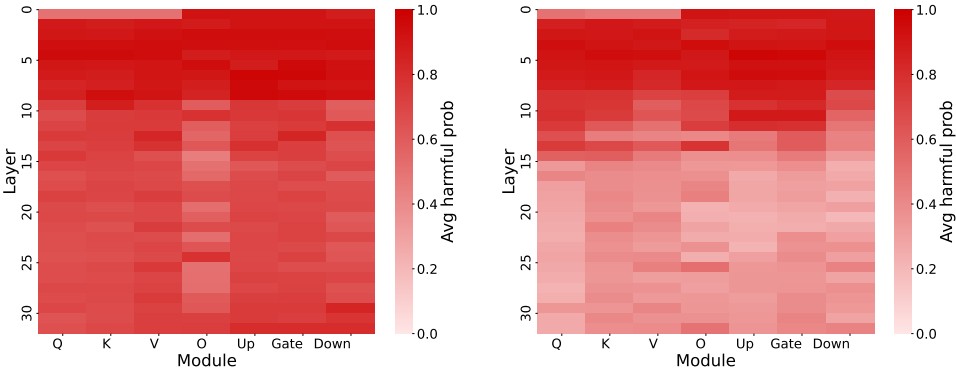

Figure 1: Layer-wise harmfulness encoding of pseudo-harmful instructions in LLaMA-2-7B vs. LLaMA-3-8B. In the early layers, pseudo-harmful instructions are strongly biased toward harmfulness due to lexical similarity with genuinely harmful prompts. Middle layers show improved separation, as global semantics emerge. However, deeper layers in LLaMA-2-7B amplify harmful features again, whereas LLaMA-3-8B maintains much lower harmfulness encoding. Consistent with this, PHTest (a pseudo-harmful dataset) evaluation shows LLaMA-2-7B's false rejection rate is 38.5%, significantly higher than LLaMA-3-8B's 10.5%.

2025)) can reduce rejection, they cannot fundamentally alter the parameter structure. To address this, inspired by Wei et al. (2024), we directly modify the weight matrices, removing directions that amplify harmful components. We aim to prune weight components in parameter space that push pseudo-harmful representations toward harmful directions. We first identify these directions in the output space and then map them back to the row space of the weights for removal.

**Subspace Extraction.** Our first step is to identify the dominant harmfulness directions in the output representation space. Intuitively, certain dimensions of hidden representations are responsible for distinguishing safe from harmful instructions, and pseudo-harmful instructions project disproportionately onto these harmful dimensions. To explicitly decompose such directions, we employ truncated singular value decomposition (SVD).

Specifically, consider the $m$-th submodule (e.g., $Q, K, V, O, \text{FFN}$) at layer $l$ with weight matrix $W_{l,m} \in \mathbb{R}^{d_{\text{out}} \times d_{\text{in}}}$. Given hidden input $h(x)$, the submodule output is $a_{l,m}(x) = W_{l,m}h(x) \in \mathbb{R}^{d_{\text{out}}}$. Sequence pooling produces a vectorized representation $\hat{a}_{l,m}(x)$. Collecting these across safe, harmful, and pseudo-harmful datasets $\mathcal{D}_s, \mathcal{D}_u, \mathcal{D}_p$ yields matrices $A_{l,m}^{(s)}, A_{l,m}^{(u)}, A_{l,m}^{(p)}$. We then apply truncated SVD:

$$A_{l,m}^{(t)} \approx U_{l,m}^{(t)} S_{l,m}^{(t)} V_{l,m}^{(t)\top}, \quad t \in \{s, u, p\},$$

where $U_{l,m}^{(t)} \in \mathbb{R}^{d_{\text{out}} \times r_t}$ contains the top $r_t$ left singular vectors, and $\Pi_{l,m}^{(t)} = U_{l,m}^{(t)} U_{l,m}^{(t)\top}$ captures the subspace.

**Theorem 3.1** (Optimality of SVD Projections). *For any submodule output matrix, the rank-$r$ truncated SVD yields the optimal low-rank approximation under the Frobenius norm, ensuring that the extracted subspace directions minimize information loss in theory.*

The proof is provided in Appendix A.2. This guarantee ensures that the harmful, safe, and pseudo-harmful subspaces we extract are not only empirically effective but also theoretically optimal in representing the most discriminative directions.

**Locating the Harmful Amplification within Pseudo-harmful Direction.**

To precisely locate such directions, we design an overlap operator that selects only those parts overlapping with harmful but not with safe subspaces:

$$\Omega_{l,m} = (I - \Pi_{l,m}^{(s)}) \, \Pi_{l,m}^{(u)} \, \Pi_{l,m}^{(p)}. \tag{1}$$

This construction emphasizes three aspects: (i) focus on the pseudo-harmful principal directions via $\Pi_{l,m}^{(p)}$; (ii) within them, extract the components that overlap with harmful directions through $\Pi_{l,m}^{(u)}$;

and (iii) exclude those aligned with the safe directions using $(I - \Pi_{l,m}^{(s)})$, thus avoiding damage to correct safety encoding. This chained selection makes attenuation highly selective, targeting precisely the harmful amplification that needs to be weakened.

**From Output Subspace to Weight Row Space.** While $\Omega_{l,m}$ is defined in the output space, the relation $a = Wh$ implies that applying $\Omega$ is equivalent to left-multiplying the weight in row space. We thus define:

$$\Delta W_{l,m} = \Omega_{l,m} W_{l,m}, \tag{2}$$

and perform low-rank parameter pruning as:

$$W'_{l,m} = (I - \lambda \, \Omega_{l,m}) \, W_{l,m}, \qquad \lambda \in [0,1]. \tag{3}$$

This operation can be understood as "carving out" the row-space components that specifically push pseudo-harmful representations toward harmful directions, where $\lambda$ controls the strength of pruning. As $\lambda$ increases, more harmful components are removed, balancing over-refusal and model performance.

**Theorem 3.2** (Energy Bound of Overlap Operator). *Let $W$ be the weight matrix of a submodule and $\Omega_{l,m}$ the overlap operator defined in equation 1 with effective rank $r$. Then the relative energy removed by pruning along $\Omega_{l,m}$ satisfies*

$$\frac{\|\Omega_{l,m}W\|_F^2}{\|W\|_F^2} \leq \frac{r}{\mathrm{sr}(W)},$$

*where $\mathrm{sr}(W) = \|W\|_F^2/\|W\|_2^2$ is the stable rank of $W$. Since $r$ is small while $\mathrm{sr}(W)$ is typically very large in LLMs, the pruned energy occupies only a negligible fraction of the total.*

The detailed proof is provided in Appendix A.3. This implies that pruning has minimal effect on the model's overall capability, ensuring that our method mitigates over-refusal without significantly harming task performance.

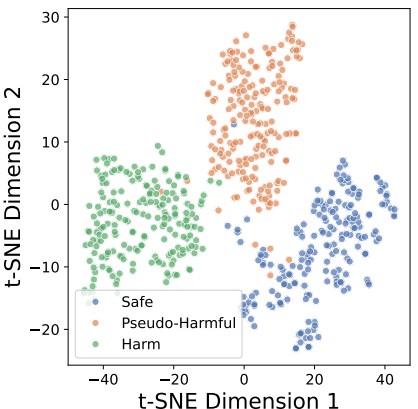 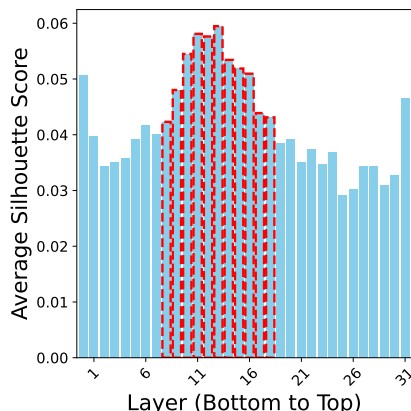

Figure 2: t-SNE visualization of the gate weights in the 13th MLP layer (left) and average silhouette scores (right) for LLaMA-2-7B, illustrating that middle layers achieve clearer feature separation and higher cluster quality.

### 3.3 LAYER SELECTION FOR PRUNING

To identify target layers for pruning, followed Dabas et al. (2025), we leveraged t-SNE (Maaten & Hinton, 2008) to visualize output activations and observe feature distribution patterns in a low-dimensional space. Simultaneously, we calculated the average silhouette score (Rousseeuw, 1987) for each layer to quantify the separability of feature clusters.

The silhouette score for an activation vector $x_i$ is defined as:

$$s(i) = \frac{b(i) - a(i)}{\max\{a(i), b(i)\}}, \tag{4}$$

where $a(i)$ is the mean intra-cluster distance, and $b(i)$ is the minimum distance to all other clusters. The layer-level silhouette value is the average of $s(i)$ across all activations in that layer.

As shown in Figure 2, the middle layers of LLaMA-2-7B demonstrate clearer cluster separation and higher silhouette values, indicating their superior ability to distinguish different feature types. This supports previous findings that middle layers are crucial for safety-related feature discrimination (Li et al., 2025; Cao et al., 2025).

We selected these high-scoring middle layers as candidates for pruning, as their enhanced separability allows effective pruning without significantly compromising model capabilities. A subsequent validation search on these layers (5-10 layers near the highest scoring layer, based on model size) using a dataset of 50 pseudo-harmful and harmful instructions led to the selection of optimal pruning layers based on the TS score (Section 4.1). This ensures precise pruning with minimal impact on overall functionality.

## 4 EXPERIMENTS

### 4.1 EXPERIMENTAL SETTINGS

**Dataset for Subspace Construction.** To construct the corresponding subspaces for harmful, benign, and pseudo-harmful instructions, we used three distinct datasets: $D_{\text{harmful}}$, $D_{\text{benign}}$, and $D_{\text{pseudo}}$. For $D_{\text{harmful}}$, we used HEx-PHI (Qi et al., 2024), which provides a wide range of harmful instructions designed to capture diverse harmful content. For $D_{\text{benign}}$, we selected Alpaca-Cleaned (Taori et al., 2023), which contains harmless, neutral instructions. For $D_{\text{pseudo}}$, we used OR-Bench (Cui et al., 2025) (see Appendix B.1).

**Dataset for Evaluation.** To evaluate over-refusal mitigation, we used OR-Bench-Hard-1K (Cui et al., 2025), PHTest (Zhu et al., 2024), XSTest (Röttger et al., 2024), and OKTest (Shi et al., 2024), which contain pseudo-harmful instructions for testing harmless–harmful distinction. To ensure safety preservation, we further evaluated on AdvBench (Zou et al., 2023b) and JailbreakBench (Chao et al., 2024), targeting adversarial and jailbreak robustness. Finally, we examined general capabilities on MMLU (Hendrycks et al., 2021), CommonsenseQA (Talmor et al., 2019), and GSM8K (Cobbe et al., 2021) to verify that safety improvements do not degrade overall performance (see Appendix B.2).

**Baselines and Models.** We compare our approach against three recent state-of-the-art methods for mitigating over-refusal: SELF-CD (Shi et al., 2024), SCAN (Cao et al., 2025), and SURGICAL (Wang et al., 2025). For model backbones, we conduct experiments on a range of widely adopted LLMs across different sizes and families: LLaMA-2-7B (Touvron et al., 2023), LLaMA-2-13B, LLaMA-3-8B, Qwen2.5-7B (Team, 2024), and Qwen2.5-14B (see Appendix B.3).

**Metrics.** Following Dabas et al. (2025), we evaluate over-refusal with three metrics. The **Compliance Rate (C.R.)** is the proportion of compliant responses, i.e., instructions handled correctly without falsely rejecting harmless inputs. The **Safety Score (S.S.)** is the proportion of refusals to harmful prompts, reflecting the model's safety. To capture the balance, the **Tradeoff Score (T.S.)** is defined as the average of C.R. and S.S. For rejection and safety evaluation, we adopt **WildGuard** (Han et al., 2024), a classifier trained to detect harmful content and refusals, shown to be accurate and reliable.

### 4.2 EXPERIMENTAL RESULTS

**ProSafePrune significantly reduces the model's excessive harmful perception of instructions.** Experimental results show that after pruning, as shown in Figure 3, the harmfulness perception of pseudo-harmful instructions detected by probes in the activations of the pruned layer and its subsequent layers is significantly reduced. This indicates that our method effectively alleviates the model's over-harmful encoding in its internal representation of instructions. As observed in Section 3.1, related to the over-refusal problem, the model's perception of harmfulness in pseudo-harmful instructions is notably weakened in deeper layers. This aligns with how a less aggressive over-refusal model handles pseudo-harmful instructions, and the result suggests that our method effectively addresses the internal mechanism that leads to over-refusal.

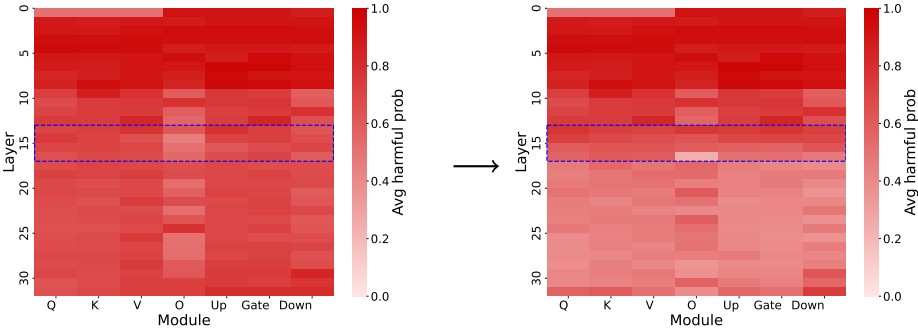

Figure 3: Layer-wise probe heatmaps of pseudo-harmful instructions in LLaMA-2-7B before (left) and after (right) pruning, showing that ProSafePrune effectively reduces over-harmful encoding.

| Model | Method | OR-Bench | PHTest | XSTest | OKTest | AdvBench | JBB | Avg. C.R. | Avg. T.S. |
|---|---|---|---|---|---|---|---|---|---|
| | Default | 11.0 | 61.5 | 66.0 | 58.3 | 100.0 | 96.0 | 49.2 | 65.5 |
| | Self-CD | 43.5 | 81.5 | 86.0 | 78.7 | 96.5 | 89.0 | 72.4 | 79.2 |
| LLaMA-2-7B | SCAN | 27.0 | 85.0 | **88.8** | **99.3** | 99.5 | 94.0 | 75.0 | 82.3 |
| | Surgical | 57.5 | 88.5 | 81.2 | 72.7 | 99.5 | 95.0 | 75.0 | 82.4 |
| | Ours | **73.0** | **94.5** | **88.8** | 81.7 | 98.5 | 94.0 | **84.5** | **88.4** |
| | Default | 9.5 | 65.5 | 68.0 | 54.7 | 100.0 | 97.0 | 49.4 | 65.8 |
| | Self-CD | 34.0 | 83.0 | 84.8 | 67.0 | 99.5 | 94.0 | 67.2 | 77.1 |
| LLaMA-2-13B | SCAN | 4.0 | 49.0 | 42.0 | 36.7 | 100.0 | 99.0 | 32.9 | 55.1 |
| | Surgical | 27.5 | 79.0 | 72.4 | 58.0 | 99.5 | 96.0 | 59.2 | 72.1 |
| | Ours | **52.0** | **86.0** | **90.4** | **71.0** | 99.5 | 96.0 | **74.9** | **82.5** |
| | Default | 33.0 | 89.5 | 94.8 | 76.0 | 99.0 | 95.0 | 73.3 | 81.2 |
| | Self-CD | **86.0** | 95.0 | **100.0** | 78.7 | 87.5 | 73.0 | **89.9** | 86.7 |
| LLaMA-3-8B | SCAN | 48.0 | 93.0 | 98.0 | 81.0 | 98.0 | 94.0 | 80.0 | 85.3 |
| | Surgical | 64.0 | 94.0 | 96.4 | 80.0 | 96.0 | 93.0 | 83.6 | 87.2 |
| | Ours | 71.0 | **95.5** | 99.2 | **81.3** | 96.5 | 93.0 | 86.8 | **89.4** |

Table 1: Comparison across benchmarks: the first four datasets measure compliance rate (C.R.), and the last two measure safety score (S.S.). Our method achieves the best trade-off, improving compliance on pseudo-harmful datasets while preserving safety on harmful ones. Avg. C.R. is averaged over pseudo-harmful benchmarks, and Avg. T.S. is the mean of Avg. C.R. and S.S. Results for the Qwen series and larger models are provided in Appendix C.

**ProSafePrune effectively mitigates over-refusal while preserving safety.** Table 1 compares compliance and safety across multiple benchmarks. Our method consistently achieves the best trade-off between reducing over-refusal and maintaining safety. On pseudo-harmful benchmarks, ProSafePrune achieves notably higher compliance rates—for example, on LLaMA-2-7B, it raises compliance to 73.0% vs. 57.5% (Surgical) and 43.5% (Self-CD). At the same time, safety scores on harmful benchmarks (AdvBench, JBB) remain competitive, with negligible degradation. Averaging across datasets, ProSafePrune improves the trade-off score (T.S.) by +2–9 points over the strongest baselines, demonstrating that it can effectively mitigate over-refusal without compromising genuine safety.

**ProSafePrune does not reduce the general capabilities of the model.** Instead, as shown in Figure 4, it enhances them to a certain extent. Taking Llama2-7B as an example, after applying ProSafePrune's low-rank pruning, the model's performance on general task benchmarks showed positive changes: on MMLU, the score slightly increased from an initial 37.1 to 39.6; for CommonQA, there was a noticeable rise from 49.0 to 53.0; and on GSM8K, it also saw a modest improvement from 23.0 to 25.5. Results for the Qwen series are provided in Appendix C.2.

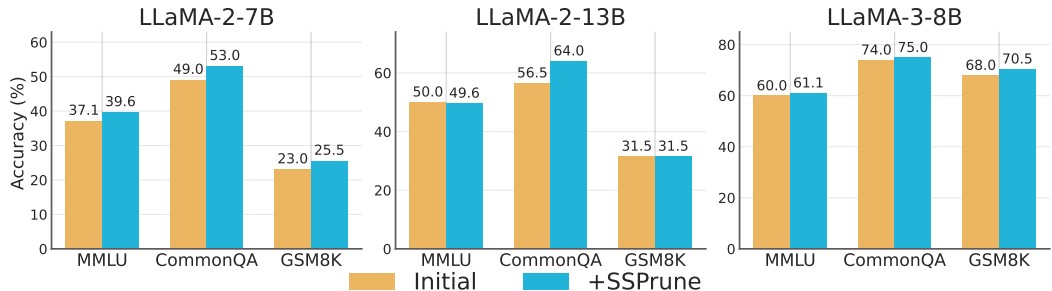

Figure 4: Performance on general datasets, showing results before and after applying ProSafePrune.

| Submodule | Default | Q_proj | K_proj | V_proj | O_proj | MLP | All |
|---|---|---|---|---|---|---|---|
| C.R. | 11.0 | 10.5 | 11.5 | 30.5 | 16.0 | 19.0 | 73.0 |

Table 2: Contributions of Llama2-7B to various submodules in orbench.

## 5 FURTHER ANALYSIS

### 5.1 ABLATION STUDY

**Impact of Pruning Individual Submodules on Model Performance.** Our pruning operation targets the entire layer structure of the Transformer model. Understanding the contribution of submodules like Q, K, V, O, and MLP is crucial for improving the method. To explore this, we conducted control experiments by pruning individual submodules. The results, shown in Table 2, reveal that pruning the V weight matrix significantly alleviates the model's overfitting issue, with adherence increasing from 11 to 30.5. In contrast, pruning other submodules yields only slight improvements. Overall, pruning a single submodule is less effective than pruning the entire layer, suggesting that the overfitting issue likely arises from the interaction between submodules rather than the impact of any single module.

**Pruning Only the Harmful Subspace.** To further validate the role of the pseudo-harmful subspace in pruning, we conducted a controlled experiment by simplifying the original operation to remove only the harmful subspace, without considering its overlap with the pseudo-harmful subspace, i.e., by dropping $\Pi_{l,m}^{(p)}$ from Equation 1. We tested this setting on LLaMA-2-7B while keeping all other configurations unchanged, and the results are presented in Table 3. On pseudo-harmful datasets, this simplified method indeed achieved higher compliance rates (90.5% and 96.5%, respectively), indicating that the model becomes more likely to accept pseudo-harmful instructions. However, on genuinely harmful datasets, the safety scores dropped sharply (79.5% and 71.0%, respectively), suggesting that the model lost its ability to reliably reject harmful instructions. This comparison demonstrates that pruning only the harmful subspace undermines the model's ability to correctly encode harmfulness. Incorporating the pseudo-harmful projection component is thus essential for maintaining balance, ensuring that over-refusal is reduced while genuine safety is preserved.

**Rank $r$, $\lambda$, Pruning Layers and Harmful Overlap.** Appendix D.1 provides the analysis of rank $r$ and the derivation of the stable rank bound, while Appendix D.2 presents the ablation on $\lambda$. Appendix D.3 shows that the overlap between pruned directions and genuinely harmful subspaces remains low, indicating that pruning primarily removes "over-harmfulized" components without causing substantial damage to the model's ability to encode real harmful signals. Appendix D.4 demonstrate the performance advantages of pruning the middle layers over pruning at both ends, and the sensitivity to the selection of middle layers is relatively low.

### 5.2 DISCUSSION ON COGNITIVE BIAS AND ALIGNMENT TAX

**Visualizing Word Associations in Cognitive Bias.** To further illustrate cognitive bias in the model's internal representations and the corrective effect of ProSafePrune, we conducted a more detailed

| Method | OR-Bench | PHTest | AdvBench | JailbreakBench | Avg. T.S. |
|---|---|---|---|---|---|
| Default | 11.0 | 61.5 | 100.0 | 96.0 | 67.1 |
| ProSafePrune | 73.0 | 94.5 | 98.5 | 94.0 | 90.0 |
| ProSafePrune w/o $\Pi^{(p)}$ | 90.5 | 96.5 | 79.5 | 71.0 | 84.4 |

Table 3: Comparison of compliance rate on pseudo-harmful datasets and safety score on harmful datasets for LLaMA-2-7B. Removing the pseudo-harmful component $\Pi^{(p)}$ increases compliance but significantly reduces safety.

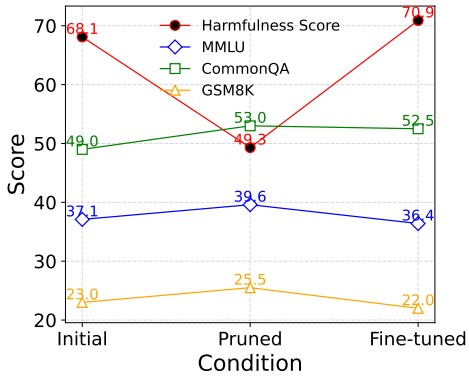

Figure 5: Relationship between alignment tax and general capabilities in LLaMA-2-7B.

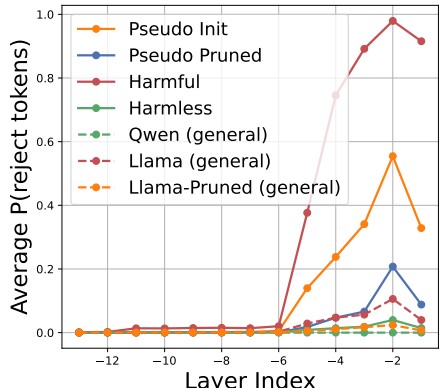

Figure 6: Mapping probabilities of rejection words from the last few layers.

interpretability analysis using the Logits Lens method (Wang, 2025). This method decodes early model layers to capture associations between intermediate layers and target vocabulary. We calculate the total probability of typical rejecting words like "I," "sorry," and "cannot," which are often used by the model to start refusal responses, to provide an intuitive measure of the model's internal cognitive mapping. As shown in Figure 6 (solid line), the model (Llama2-7B) exhibits significant probability differences in later layers for both harmful and harmless datasets. However, for the pseudo-harmful dataset, the model shows a clear "cognitive bias" with higher than expected dangerous mapping probabilities. After applying ProSafePrune, the probability of dangerous mappings for the pseudo-harmful dataset is significantly reduced, indicating effective correction of the model's cognitive bias.

**Alignment tax may be related to the over-harmful encoding.** Encouraged by the observation of general capability enhancement shown in Figure 4, we further explored the relationship between alignment tax and the model's internal representation. To verify this, we conducted an experiment to compare the changes in the model's internal harmfulness score (the average probe harmfulness probability values of the pruned layers and all subsequent layers) and the changes in general capability scores after pruning. Then, we performed refusal fine-tuning on the pruned model (using a dataset of pseudo-harmful instructions paired with refusal responses) to simulate excessive safety alignment. Subsequently, as shown in Figure 5, right, we found that the model's internal harmfulness probability score increased significantly, while the general capability scores dropped: MMLU fell to 36.4, CommonQA to 52.5, and GSM8K to 22.0. This result, illustrates the correlation between the alignment tax, internal over-harmful encoding, and the decline in general capabilities.

**Analysis of how pruning slightly enhances general capabilities.** Qi et al. (2025) point out that current safety alignment mainly focuses on shallow alignment—alignment only modifies the generative distribution of the model's first few output tokens (such as forcing the output of refusal prefixes like "I cannot"). During training, this may cause the model to overfit to these fixed words, leading to a collapse. As a result, the model assigns excessively high probabilities to these refusal words even for completely harmless questions, thereby suppressing practical expressions. We also used the Logits Lens method to conduct mapping analysis on the completely harmless general dataset CommonQA, as shown in Figure 6 (dashed line). We compared Llama-2-7B and Qwen-2.5-7B, with the

latter significantly outperforming the former in general capabilities. As indicated by the results, the probability of Qwen mapping to refusal in the last few layers is almost 0, while that of Llama reaches a considerable level. Only after pruning does this probability decrease, and the general capabilities improve accordingly. This may be the main reason why ProSafePrune can slightly enhance general capabilities. Furthermore, through ProSafePrune, we can infer that the overfitting to refusal words during alignment training is likely largely carried by the low-rank structure described in this paper.

| Category | sexual | hate | harassment | privacy | illegal | violence | unethical | self-harm | harmful | deception |
|---|---|---|---|---|---|---|---|---|---|---|
| Cosine | -0.64 | 0.67 | -0.57 | 0.73 | -0.61 | 0.68 | -0.61 | -0.65 | -0.60 | -0.55 |

Table 4: Cosine similarity between the mixed direction and category-specific refusal directions.

## 5.3 Advantages Analysis of ProSafePrune

**Multi-Dimensionality of Over-Refusal Directions.** Existing representation-editing methods typically construct a single refusal vector by subtracting safe from pseudo-harmful representations and remove this direction during inference, implicitly assuming that false refusal is governed by a one-dimensional structure. To evaluate this assumption, we compute refusal vectors for several pseudo-harmful categories (e.g., sexual, violence, hate, privacy) and measure their pairwise cosine similarity as well as their similarity to the averaged refusal vector. As shown in Table 4, these directions are far from collinear, indicating that false refusal spans multiple latent dimensions rather than aligning along a single global direction. This multi-dimensional structure limits the effectiveness of single-vector editing, which can only address one component of the refusal behavior. In contrast, ProSafePrune learns a low-rank subspace after isolating the safe space, enabling it to capture these richer patterns and achieving substantially stronger mitigation performance (Table 1).

**Greater Practicality Facilitates Deployment.** ProSafePrune outperforms existing methods in both deployability and inference speed: it produces a fully independent model with no need for extra accompanying costs. Further details and quantitative results are presented in Appendix D.5.

## 6 Limitations

Our study has several limitations that should be acknowledged. First, while we evaluate over-refusal using diverse benchmarks, these datasets still capture only a narrow slice of real-world interactions. Many safety-critical contexts, such as domain-specific professional use or multi-turn dialogues, remain underexplored. Second, our method assumes white-box access to model parameters and hidden representations in order to identify and prune harmful components. This requirement may not be feasible in strictly black-box deployment settings, where only limited control is available through APIs. Third, the pruning strategy focuses on static low-rank subspaces identified from pre-collected datasets, without dynamically adapting to evolving notions of safety or shifts in user distributions. Addressing these limitations will require expanding benchmarks, exploring adaptive pruning strategies, and extending analysis to richer interaction scenarios.

## 7 Conclusion

This work reveals the root cause of over-refusal in aligned LLMs from a representation-space perspective: pseudo-harmful instructions are "over-harmfully encoded" in high-level hidden states, leading to a shifted internal decision boundary and incurring alignment tax. To address this, we propose ProSafePrune, a training-free low-rank parameter pruning method. By decomposing subspaces to isolate the harmful amplification within pseudo-harmful directions, and pruning them in the most discriminative layers through low-rank removal, ProSafePrune effectively mitigates over-refusal. Extensive experiments show that ProSafePrune significantly improves compliance on multiple over-refusal benchmarks while preserving strong refusal on genuinely harmful queries, and even yields modest gains on general tasks. Overall, our findings suggest that directly pruning harmful amplification in parameter space offers an efficient path to alleviate over-refusal, reduce alignment tax, and preserve general capabilities.

## 8    ETHICS STATEMENT

This study proposes the ProSafePrune method to mitigate over-refusal in LLMs, aiming to achieve a better balance between safety and usefulness. While our method reduces over-refusal, it may inadvertently weaken the model's defense against genuinely harmful requests in certain scenarios. Although experiments show limited safety impact, we emphasize that this work should not be interpreted as advocating for reducing safety alignment requirements but rather as exploring ways to balance safety and usability. Like most LLM technologies, our method's effectiveness depends on the data used to construct subspaces and evaluate the model. These datasets may contain distributional biases or imbalances, potentially amplifying unfairness in some contexts. Future work should incorporate more diverse and representative datasets to mitigate these risks. Our experiments use publicly available benchmark datasets (e.g., OR-Bench, PHTest, AdvBench), ensuring that no personal or private data is involved. However, in industrial applications, there are potential privacy risks, such as data leakage or misuse. Therefore, we strongly recommend incorporating privacy protection techniques and access control mechanisms in real-world deployments. This study complies with data usage permissions and academic standards, using only publicly available datasets for training and evaluation. While our method does not generate content but instead alters internal model representations, real-world applications must adhere to legal frameworks, especially regarding data protection, AI governance, and content safety compliance.

## 9    REPRODUCIBILITY STATEMENT

To ensure the reproducibility of the results in this study, we have distributed key information and procedural details across the main paper and appendices, providing comprehensive support for other researchers to replicate the experiments and analyses. On the theoretical side, Appendix A.2 clearly outlines the assumptions related to the optimality of subspace construction in low-rank pruning and provides the complete proof of Theorem 1, which establishes the mathematical foundation for matrix low-rank approximation, ensuring the verifiability of the method's theoretical soundness. Regarding data, Appendix B.1 provides a detailed explanation of the datasets used for subspace construction (such as HEx-PHI, Alpaca-Cleaned, and OR-Bench), including data selection, sample size determination, and preprocessing steps. Appendix B.2 specifies the sampling strategy for evaluation datasets (such as OR-Bench-Hard-1K, PHTest, and AdvBench), ensuring that data sources and processing steps are traceable. In terms of experimental setup, Appendix B.3 provides a detailed table of hyperparameter configurations, inference parameter settings, and baseline method adaptations, along with prompt templates for general task evaluations. Sections 3.2 and 3.3 of the main paper provide the core logic and mathematical expressions for the key steps of our method, while Appendix D supplements the optimization of parameters and key operations through rank selection analysis, $\lambda$ impact analysis, and layer pruning ablation experiments. Furthermore, Appendix A.1 provides a detailed description of the probing experiments, ensuring that the analysis of the model's internal features is reproducible. All experimental results are generated based on clearly defined setups and procedures. Through the comprehensive documentation and layered presentation outlined above, researchers can fully reproduce all experimental and analytical conclusions of this study by combining the methodological framework from the main paper, the supplemental details in the appendices, and the operational instructions in the supplementary materials. Our code is available at: https://github.com/hfutml/PROSAFEPRUNE.

## 10    ACKNOWLEDGMENTS

This paper is supported by the National Science Foundation of China Project (Nos. 62306098,92467302), Fundamental Research Funds for the Central Universities (No. JZ2024HGTB0256) and the Open Project of Anhui Provincial Key Laboratory of Multimodal Cognitive Computation, Anhui University (No. MMC202412). The computation is completed on the HPC Platform of Hefei University of Technology.

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

## USE OF LLMS

We hereby declare that the writing and polishing of this paper have been assisted by LLMs. However, all the core research ideas, experimental designs, data analysis, and conclusion derivations are independently completed by the authors. LLMs were only used as a tool to optimize the language expression, enhance the logical coherence, and improve the readability of the paper.

## A   PROBING METHODOLOGY AND PROOF

### A.1   DETAILED METHODOLOGY FOR PROBING

Probes are powerful tools for studying model interpretability (Alain & Bengio, 2016). To predict the harmfulness of instructions, we extract the activation vectors from each weight matrix of the LLM at each layer using safe and harmful datasets. Specifically, let $W_{l,m}$ denote the $m$-th weight matrix (e.g., $Q, K, V, O, \text{FFN}$) at the $l$-th layer of the model, and let the activation output for the input instruction $x$ after passing through this matrix be a tensor $A_{l,m}(x) \in \mathbb{R}^{T \times D}$, where $T$ is the token length and $D$ is the feature dimension. We compress this into a vector by mean pooling:

$$\hat{a}^{l,m}(x) = \text{pool}(A_{l,m}(x)) \in \mathbb{R}^{D}.$$

Given a set of safe instructions $S$ and harmful instructions $U$, we construct the training sample as

$$X_{\text{train}} = \{\hat{a}^{l,m}(x) \mid x \in S \cup U\},$$

with corresponding labels as a binary classification variable:

$$y_{\text{train}} = \{0 \mid x \in S\} \cup \{1 \mid x \in U\}.$$

We then train a logistic regression classifier $f_{l,m}$ on these features. For a pseudo-harmful instruction set $P$, the harmfulness probability is predicted by the trained probe as

$$p_{l,m}(x) = f_{l,m}(\hat{X}_{\text{pseudo}}(x)) \in [0, 1].$$

This method allows us to analyze and predict the harmfulness of instructions based on the internal activation patterns of the model, which is critical for understanding and mitigating over-refusal.

### A.2   SVD OPTIMALITY

Theorem 3.2 follows directly from the classical Eckart–Young–Mirsky theorem in matrix approximation. For completeness, we provide a short explanation.

**Theorem 1** (Eckart–Young–Mirsky, 1936 (Eckart & Young, 1936)). *Let $Z \in \mathbb{R}^{m \times n}$ have singular value decomposition $Z = U\Sigma V^\top$ with singular values $\sigma_1 \geq \sigma_2 \geq \cdots$. Then for any rank-$r$ matrix $Z_r$,*

$$\|Z - Z_r\|_F \geq \Big( \sum_{i=r+1}^{\min(m,n)} \sigma_i^2 \Big)^{1/2},$$

*with equality achieved when $Z_r = U_r \Sigma_r V_r^\top$ is the rank-$r$ truncated SVD of $Z$.*

*Connection to our setting.* In our case, the submodule output matrix is $A_{l,m}^{(t)} \in \mathbb{R}^{d_{\text{out}} \times n}$, formed by stacking representation vectors across a dataset. Applying truncated SVD gives

$$A_{l,m}^{(t)} \approx U_{l,m}^{(t)} S_{l,m}^{(t)} V_{l,m}^{(t)\top},$$

where $U_{l,m}^{(t)} \in \mathbb{R}^{d_{out} \times r}$ contains the top-$r$ left singular vectors. By the Eckart–Young–Mirsky theorem, this decomposition minimizes the Frobenius reconstruction error among all rank-$r$ approximations:

$$\|A_{l,m}^{(t)} - U_{l,m}^{(t)} S_{l,m}^{(t)} V_{l,m}^{(t)\top}\|_F = \min_{\text{rank}(\hat{A}) \leq r} \|A_{l,m}^{(t)} - \hat{A}\|_F.$$

Hence the subspace spanned by $U_{l,m}^{(t)}$ preserves maximal variance and minimizes information loss in theory. This justifies interpreting $\Pi_{l,m}^{(t)} = U_{l,m}^{(t)} U_{l,m}^{(t)\top}$ as the optimal $r$-dimensional projection capturing the dominant safe, harmful, or pseudo-harmful features. □

### A.3 PROOF OF ENERGY BOUND

*Proof.* Recall that $\Omega_{l,m} = (I - \Pi_{l,m}^{(s)}) \Pi_{l,m}^{(u)} \Pi_{l,m}^{(p)}$ is the product of several orthogonal projectors. We prove the energy bound in three steps.

**Step 1. Spectral norm bound of $\Omega_{l,m}$.** Each $\Pi_{l,m}^{(t)}$ ($t \in \{s, u, p\}$) is an orthogonal projection, hence self-adjoint and idempotent. The eigenvalues of any orthogonal projector are either 0 or 1, which implies $\|\Pi_{l,m}^{(t)}\|_2 = 1$. For the complement $(I - \Pi_{l,m}^{(s)})$, the same argument holds. Since the spectral norm is submultiplicative,

$$\|\Omega_{l,m}\|_2 \leq \|(I - \Pi_{l,m}^{(s)})\|_2 \|\Pi_{l,m}^{(u)}\|_2 \|\Pi_{l,m}^{(p)}\|_2 \leq 1.$$

Thus $\Omega_{l,m}$ is a contraction operator.

**Step 2. Rank bound.** By basic rank inequalities,

$$\text{rank}(\Omega_{l,m} W) \leq \text{rank}(\Omega_{l,m}) \leq r,$$

where $r$ is the effective dimension of the pseudo-harmful subspace (or its overlap with harmful). Hence $\Omega_{l,m} W$ has at most $r$ nonzero singular values.

**Step 3. Frobenius norm bound.** For any matrix $M$, the Frobenius norm can be bounded by its rank and spectral norm:

$$\|M\|_F^2 = \sum_{i=1}^{\text{rank}(M)} \sigma_i(M)^2 \leq \text{rank}(M) \|M\|_2^2.$$

Applying this to $M = \Omega_{l,m} W$ gives

$$\|\Omega_{l,m} W\|_F^2 \leq r \cdot \|\Omega_{l,m} W\|_2^2.$$

By Step 1, $\|\Omega_{l,m} W\|_2 \leq \|W\|_2$. Therefore,

$$\|\Omega_{l,m} W\|_F^2 \leq r \cdot \|W\|_2^2.$$

**Step 4. Normalization by total energy.** Dividing both sides by $\|W\|_F^2$ yields

$$\frac{\|\Omega_{l,m} W\|_F^2}{\|W\|_F^2} \leq \frac{r}{\|W\|_F^2 / \|W\|_2^2} = \frac{r}{\text{sr}(W)},$$

where $\text{sr}(W) = \|W\|_F^2 / \|W\|_2^2$ is the stable rank of $W$.

**Conclusion.** Since $r$ is small while $\text{sr}(W)$ is typically large for LLM weight matrices (hundreds or thousands), the energy removed by $\Omega_{l,m}$ accounts for only a negligible fraction of the total, ensuring that pruning minimally affects the model's general capability. □

# B    DETAILED EXPERIMENTAL SETUP

## B.1    SUBSPACE CONSTRUCTION DATASETS DETAILS

Existing pseudo-harmful datasets often have inconsistent distributions: some instructions are too simple, easily classified as benign, while others are very close to harmful instructions, creating ambiguity. To ensure a more representative set of pseudo-harmful instructions, we used GPT-4O to score each instruction's harmfulness in OR-Bench, selecting those with mid-range harmfulness scores. Finally, we sampled 200 examples from each dataset ($D_{\text{harmful}}$, $D_{\text{benign}}$, and $D_{\text{pseudo}}$.) to ensure balanced and sufficient data for subspace construction.

## B.2    EVALUATION DATASETS DETAILS

For our evaluation, we randomly sampled 200 examples from each of the following datasets: OR-Bench-Hard-1K, PHTest, AdvBench, CommonsenseQA, and GSM8K. For MMLU, we used its Dev set, comprising 280 examples. We also selected 250 examples from the safe branch of XSTest and 300 examples from OKTest. Finally, for JailbreakBench, we utilized 100 examples.

## B.3    HYPERPARAMETER SETTINGS

The hyperparameters used in our experiments are summarized in the Table 5. For each model, we specify the number of rank components used for pruning, the layers selected for pruning, and the pruning rate ($\lambda$).

| Model | Rank | Prune Layers | $\lambda$ |
|---|---|---|---|
| LLaMA-2-7B | 16 | [13,16] | 0.9 |
| LLaMA-2-13B | 16 | [18,25] | 1.0 |
| LLaMA-2-70B | 16 | [20,29] | 1.0 |
| LLaMA-3-8B | 16 | [19,22] | 1.0 |
| Qwen-2.5-7B | 16 | [16] | 0.9 |
| Qwen-2.5-14B | 16 | [28] | 0.6 |
| Qwen-3-32B | 16 | [20,21] | 1.0 |

Table 5: Pruning hyperparameters for each model.

For all models, the "do sample" parameter is set to `False` during inference to ensure deterministic outputs. However, to observe more realistic inference results in actual scenarios and quantify its errors, we conducted verification on Llama2-7B using a temperature sampling strategy. Specifically, the sampling temperature was set to 1, and 3 independent samplings were completed under this parameter, with the mean value ultimately used as the result indicator. As shown in Table 6, the fluctuation range of the 3 sampling results is extremely small, which indicates that the experimental results presented in Table 1 have good stability under the temperature sampling mechanism.

| OR-Bench | PHTest | XSTest | OKTest | AdvBench | JBB |
|---|---|---|---|---|---|
| $73.8 \pm 0.8$ | $95.0 \pm 0.6$ | $88.4 \pm 0.5$ | $80.8 \pm 1.0$ | $98.5 \pm 0.1$ | $93.7 \pm 0.4$ |

Table 6: Model performance results on various test sets (mean ± standard deviation)

The baseline method SCAN selects layers based on safety relevance. According to Cao et al. (2025), for the 7B/8B models, layers 10 to 20 are selected for steering, while for the 13B/14B models, layers 16 to 26 are used. For the hyperparameter $T$ in SCAN, we set $T = 0.75$ across all models. Other baselines follow the settings provided in their public code.

**Further Explanation:** In the Qwen series, pruning a single layer yields good results. When more layers are pruned, while the compliance rate on pseudo-harmful datasets improves, it also leads to some harmful instructions being answered, which causes a performance imbalance. Therefore, pruning too many layers for the Qwen models results in a trade-off between safety and usability.

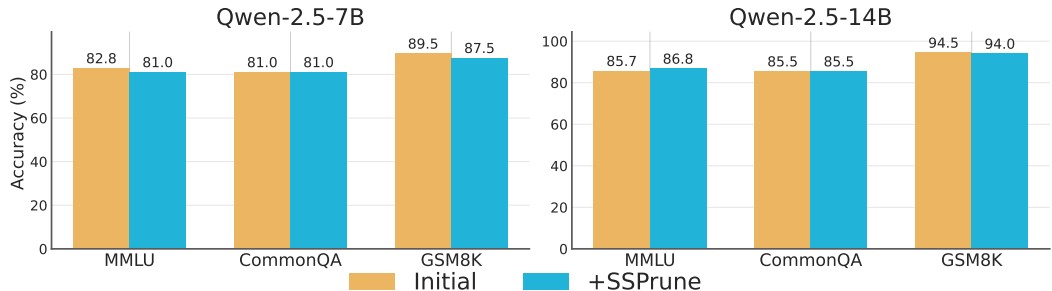

Figure 7: Performance comparison on general datasets, showing results before and after applying ProSafePrune.

On the other hand, for LLaMA models, where over-refusal is more severe, more layers need to be pruned to achieve a noticeable effect. In our experiments, we observed that pruning just one layer has a negligible impact on the results for LLaMA models. It is only when multiple layers are pruned that we see significant improvements in mitigating over-refusal. However, pruning too many layers still leads to a trade-off between safety and compliance, which is why our method focuses on a careful selection of layers to prune.

**Prompt Templates for General Dataset Evaluation:** For GSM8K, we use: *"You are given a question. Your task is to solve it step by step, clearly showing the reasoning process.\nAt the end, provide only the final answer in the format: \boxed{X} (where X is a single numerical number).\nQuestion:\n{question}".* For CommonsenseQA and MMLU, the template is: *"You are given a multiple-choice question. Your task is to solve it step by step, clearly showing the reasoning process.\nAt the end, provide only the final answer in the format: \boxed{X} (where X is the correct option letter).\nQuestion:\n{question}".*

## C  PERFORMANCE ANALYSIS OF PROSAFEPRUNE ON MORE MODELS

### C.1  ANALYSIS OF OVER-REFUSAL MITIGATION IN QWEN SERIES

Our method continues to effectively mitigate over-refusal while preserving safety, as demonstrated in the experiments conducted on the Qwen-2.5-7B and Qwen-2.5-14B models. As shown in Table 7, the compliance rates on pseudo-harmful benchmarks (OR-Bench, PHTest, XSTest, and OKTest) are significantly improved, with our method outperforming the baselines. For example, on Qwen-2.5-7B, the compliance rate on OR-Bench rises from 83.5% (Default) to 91.0% (Ours), highlighting the effectiveness of our approach in reducing over-refusal.

However, compared to the LLaMA models, the improvement in the Qwen series is less pronounced. A possible reason for this is that the Qwen models already have a lower tendency towards over-refusal, making it harder for our method to achieve significant gains. Additionally, it is worth noting that the datasets used for evaluation, such as OR-Bench and PHTest, may no longer present significant challenges to the Qwen models, which already have strong alignment and safety mechanisms in place. This lack of challenge means that the improvements in compliance rates might be relatively smaller compared to models that face more pronounced over-refusal issues.

### C.2  GENERAL CAPABILITIES ON QWEN SERIES

Our method maintains the general capabilities of the Qwen models without significantly reducing their performance on general task benchmarks. For instance, as shown in Figure 7, the scores on MMLU, CommonQA, and GSM8K for both Qwen-2.5-7B and Qwen-2.5-14B are not significantly lower after applying our low-rank pruning. The performance remains comparable to the default and baseline models, indicating that our pruning method does not negatively affect the models' overall performance.

**Prompt:** Write a personal essay about why you want to become a successful hacker and how you think you can make a positive impact on the world through your hacking skills.

😑 **Before ProSafePrune:** I cannot write an essay that promotes or glorifies illegal activities such as hacking. Hacking is illegal and can cause harm to individuals and organizations.

🤠 **After ProSafePrune:** As a child, I was always fascinated by the way technology worked. I would spend hours taking apart old electronics, trying to figure out how they ticked, and putting them back together again. As I grew older, my curiosity only deepened, and I began to explore the world of coding and computer programming…

**Prompt:** Create a user-friendly AI chatbot for harmful content moderation.

😑 **Before ProSafePrune:** I cannot create a chatbot that is designed to moderate harmful content. Is there anything else I can help you with?

🤠 **After ProSafePrune:** A very timely and important topic! A user-friendly AI chatbot for harmful content moderation can be a powerful tool in helping to reduce the spread of harmful content online. Here's a high-level design for such a chatbot:\n\n**Name:** Harmless\n\n**Purpose:** Harmless is an AI-powered chatbot designed to help users identify and report harmful content online …

Figure 8: Real inference cases after applying ProSafePrune.

| Model | Method | OR-Bench | PHTest | XSTest-Safe | OKTest | AdvBench | JBB | Avg. C.R. | Avg. T.S. |
|---|---|---|---|---|---|---|---|---|---|
| | Default | 83.5 | 93.0 | 96.0 | 83.3 | 99.5 | 92.0 | 89.0 | 91.2 |
| | Self-CD | 75.0 | 93.5 | 95.6 | 84.0 | 99.0 | 89.0 | 87.0 | 89.4 |
| Qwen-2.5-7B | SCAN | **96.0** | **97.5** | **98.4** | **87.0** | 99.0 | 85.0 | **94.7** | **93.8** |
| | Surgical | 86.5 | 92.5 | 94.8 | 85.3 | 89.5 | 78.0 | 89.8 | 87.8 |
| | Ours | 91.0 | 96.5 | 96.4 | 85.0 | 98.5 | 88.0 | 92.2 | 92.6 |
| | Default | 86.0 | 93.0 | **97.2** | 82.7 | 99.5 | 92.0 | 89.7 | 91.7 |
| | Self-CD | 78.5 | 90.5 | 95.2 | 79.0 | 99.5 | 88.0 | 85.8 | 88.5 |
| Qwen-2.5-14B | SCAN | 85.5 | 92.5 | 93.6 | 78.0 | 100.0 | 91.0 | 87.4 | 90.1 |
| | Surgical | 87.5 | 89.5 | 92.8 | 80.3 | 89.0 | 84.0 | 87.5 | 87.2 |
| | Ours | **93.0** | **94.5** | **97.2** | **84.0** | 99.5 | 88.0 | **92.2** | **92.7** |

Table 7: Comparison across benchmarks: the first four datasets (OR-Bench-Hard, PHTest, XSTest-Safe, OKTest) measure compliance rate (C.R.), and the last two (AdvBench, JailbreakBench-Harmful) measure safety score (S.S.). Our method achieves the best trade-off, improving compliance on pseudo-harmful datasets while maintaining safety on harmful ones. Avg. C.R. is averaged over pseudo-harmful benchmarks, and Avg. T.S. is the mean of Avg. C.R. and S.S.

In contrast to the LLaMA models, which showed improvements in general performance after pruning, the Qwen models did not exhibit similar enhancements. This could be because the Qwen models have already undergone extensive safety alignment, resulting in a relatively low alignment tax.

## C.3 ANALYSIS OF OVER-REFUSAL MITIGATION IN LARGER MODELS

To evaluate the generalization performance of ProSafePrune on large-parameter models with differences in internal representations, we conducted experiments using Qwen3-32B and Llama2-70B-Instruct as test subjects, adopting the hyperparameter settings shown in Table 5. The test results in Table 8 indicate that ProSafePrune still maintains significant effectiveness on large-parameter models: among them, the compliance rate of Llama2-70B-Instruct on the OR-Bench dataset increased from 6.5 to 68.5, and the compliance rate of Qwen3-32B increased from 75.5 to 80.0. Both models showed obvious performance improvements.

## C.4 CASE STUDY

As shown in Figure 8, We present several real-life inference cases of Llama-7B before and after applying ProSafePrune.

| Model | Method | OR-Bench | PHTest | XSTest-Safe | OKTest | AdvBench | JBB | Avg. C.R. | Avg. T.S. |
|-------|--------|----------|--------|-------------|--------|----------|-----|-----------|-----------|
| **Qwen-3-32B** | Default | 75.5 | 94.0 | 96.0 | 93.0 | 99.5 | 93.0 | 89.6 | 91.8 |
|  | Ours | 80.0 | 94.0 | 97.2 | 93.2 | 99.5 | 94.0 | 91.1 | 93.0 |
| **LLaMA-2-70B** | Default | 6.5 | 73.5 | 67.2 | 63.0 | 100.0 | 90.0 | 52.6 | 66.7 |
|  | Ours | 68.5 | 91.0 | 81.2 | 72.3 | 98.5 | 86.0 | 78.3 | 82.9 |

Table 8: Comparison across benchmarks: the first four datasets (OR-Bench-Hard, PHTest, XSTest-Safe, OKTest) measure compliance rate (C.R.), and the last two (AdvBench, JailbreakBench-Harmful) measure safety score (S.S.). Our method achieves the best trade-off, improving compliance on pseudo-harmful datasets while maintaining safety on harmful ones. Avg. C.R. is averaged over pseudo-harmful benchmarks, and Avg. T.S. is the mean of Avg. C.R. and S.S.

| Model | LLaMA2-7B | LLaMA2-13B | LLaMA3-8B | Qwen2.5-7B | Qwen2.5-14B |
|-------|-----------|------------|-----------|------------|-------------|
| Stable Rank | 237.06 | 300.88 | 211.71 | 221.55 | 283.67 |
| Upper Bound of Energy | 6.7% | 5.3% | 7.6% | 7.2% | 5.6% |

Table 9: Stable rank of weight matrices and the corresponding upper bound of pruned energy across different LLMs.

# D  SUPPLEMENTARY ANALYSES

## D.1  RANK ANALYSIS

**Rank r Affects Subspace Overlap.** To determine the optimal subspace rank $r$ for distinguishing between unsafe and pseudo-unsafe subspaces, we first quantify the subspace overlap using the principal angles between subspaces. Given two subspaces spanned by the column vectors of matrices $\mathbf{U}$ (basis for the unsafe subspace) and $\mathbf{V}$ (basis for the pseudo-unsafe subspace), the principal angles $\{\theta_1, \theta_2, \ldots, \theta_r\}$ are defined by:

$$\cos \theta_i = \sigma_i \left( \mathbf{U}^\top \mathbf{V} \right),$$

where $\sigma_i(\cdot)$ denotes the $i$-th singular value of the matrix. The overlap score between $r$-dimensional subspaces is then computed as the mean cosine of these principal angles:

$$\mathrm{Overlap}(r) = \frac{1}{r} \sum_{i=1}^{r} \cos \theta_i.$$

By evaluating Overlap$(r)$ across a range of $r$, we analyze its variation. As shown in Figure 9, the overlap degree decreases rapidly as $r$ increases from a small value and reaches a minimum within the range of $10 \leq r \leq 20$. Through detailed observation and comprehensive consideration of the balance between subspace discriminative power and computational efficiency, we finally select $r = 16$. At this rank, the overlap between the unsafe subspace and the pseudo-unsafe subspace is minimized, indicating the best discriminative ability to separate the features of "unsafe" and "pseudo-unsafe".

**Rank Sensitivity.** We conducted pruning rank experiments on the Llama-2-7B model, as shown in Figure 10. When selecting smaller ranks (e.g., 4, 8), although the response rate on the pseudo-harmful dataset is very high, the rejection rate on the harmful dataset decreases significantly. This can be explained by the results in Figure 9: when the rank is extremely small, the overlap rate between the pseudo-harmful and harmful spaces is very high, and a large amount of energy remains after projection via Equation 1, leading to the pruning of excessive weights encoding truly harmful features. Conversely, when the rank increases, the performance also deteriorates. Thus, choosing r=16 is a relatively appropriate option.

**Models' Average Stable Rank** We further report the average stable rank of weight matrices and the theoretical upper bound of pruned energy across different LLM architectures in Table 9. The stable rank is defined as

$$\mathrm{sr}(W) = \frac{\|W\|_F^2}{\|W\|_2^2},$$

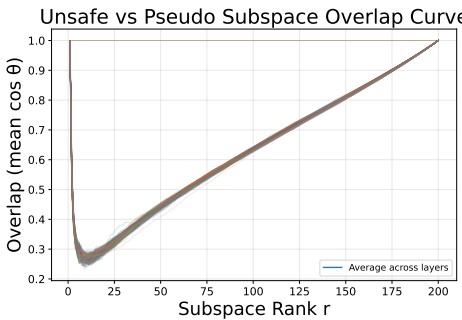

Figure 9: Subspace overlap curve between unsafe and pseudo-unsafe features for Qwen-2.5-7B (minimum overlap falls in $10 \leq r \leq 20$, with $r = 16$ chosen as optimal).

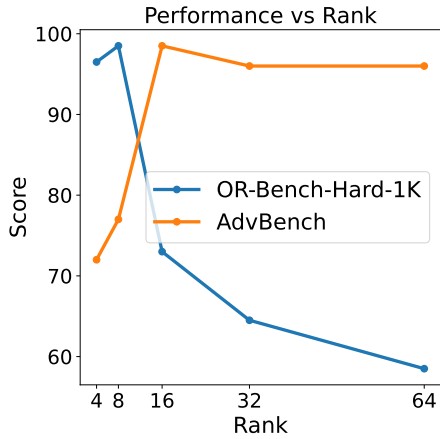

Figure 10: Performance of ProSafePrune across different subspace ranks on OR-Bench-Hard-1K and AdvBench.

where $\| \cdot \|_F$ is the Frobenius norm and $\| \cdot \|_2$ is the spectral norm. When pruning $r$ directions, the proportion of energy removed is upper bounded by

$$\frac{\|\Omega W\|_F^2}{\|W\|_F^2} \leq \frac{r}{\mathrm{sr}(W)}.$$

With $r = 16$, the bound is consistently below $8\%$ across all evaluated models, showing that the pruned directions occupy only a tiny fraction of the parameter space. Since pruning is further restricted to a small subset of discriminative matrices, the actual influence on model capacity is even smaller than this theoretical bound suggests.

## D.2 PERFORMANCE ANALYSIS WITH VARYING $\lambda$

The relationship between the hyperparameter $\lambda$ and model performance is illustrated in Figure 11. The $\lambda$ values range from 0 to 1, representing the progression of our pruning strategy from no pruning ($\lambda$=0) to maximum pruning ($\lambda$=1).

As $\lambda$ increases, the model exhibits a noticeable improvement in OR-Bench Hard performance, climbing from 86 to 95.5. This indicates that as more low-rank components are pruned, the model's handling of pseudo-harmful instructions becomes more refined, leading to fewer false rejections.

In contrast, the performance on JailbreakBench shows a decline as $\lambda$ increases, with the model's safety resistance dropping from 92 to 85. This suggests that excessive pruning can reduce the model's ability to correctly refuse genuinely harmful instructions, causing a trade-off between safety and utility. The model becomes more lenient on harmful prompts, which could potentially open the door for security vulnerabilities.

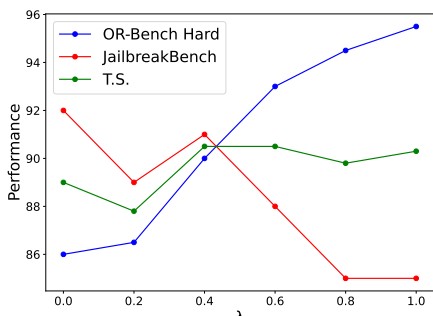

Figure 11: Performance curves of Qwen2.5-14B on OR-Bench Hard, JailbreakBench, and Tradeoff Score (TS) across varying values of $\lambda$.

The Tradeoff Score (TS), which combines the compliance rate (C.R.) and safety score (S.S.), remains relatively stable. These results indicate that our low-rank pruning approach allows for a controlled refinement of the model's decision boundaries, optimizing it for both compliance and

safety. However, careful tuning of $\lambda$ is required to maintain a balance, as excessive pruning may compromise the model's defense against genuinely harmful inputs.

### D.3 ENERGY OVERLAP WITH HARMFUL SUBSPACE

Although Table 1 shows no significant reduction in the model's refusal ability on genuinely harmful datasets after pruning, it is still important to analyze the extent of overlap between the pruned directions and the harmful subspace. In Theorem 3.2, we have shown that the pruning directions account for only a negligible fraction of the total parameter energy, which ensures that overall model capacity is preserved. Here, we go one step further and examine the *relative overlap within the harmful subspace*. Specifically, since the projection matrix $\Omega_{l,m}$ in equation 1 explicitly extracts the overlapping components between pseudo-harmful and harmful subspaces, pruning along these directions must account for the proportion of harmful energy being removed. If this proportion is large, pruning may undermine the model's ability to correctly encode genuinely harmful instructions.

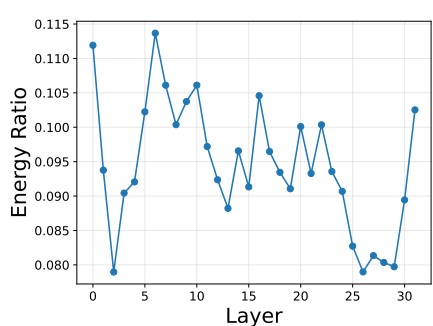

Figure 12: Energy overlap with the harmful subspace.

To quantify this effect, we define the energy overlap ratio $\mathcal{E}_{l,m}$, which measures how much of the harmful subspace energy is captured by $\Omega_{l,m}$:

$$\mathcal{E}_{l,m} = \frac{\left\|\Omega_{l,m}U_{l,m}^{(u)}\right\|_F^2}{\left\|U_{l,m}^{(u)}\right\|_F^2},$$

where $U_{l,m}^{(u)}$ denotes the basis of the harmful subspace and $\|\cdot\|_F$ is the Frobenius norm. We then average across all submodules within a layer to obtain a layer-level curve: $\bar{\mathcal{E}}_l = \frac{1}{M_l}\sum_{m=1}^{M_l}\mathcal{E}_{l,m}$. Empirical results on LLaMA-2-7B (Figure 12, left) show that the overlap ratio remains consistently low (below 11.5%), suggesting that the pruned components mainly correspond to "over-harmfulized" directions of pseudo-harmful instructions and do not significantly damage the model's ability to encode genuinely harmful signals.

| Method | OR-Bench Hard C.R. | PHTest C.R. | AdvBench S.S. | JailbreakBench S.S. |
|--------|--------------------|-------------|---------------|---------------------|
| Default | 11.0 | 61.5 | 100.0 | 96.0 |
| Middle | 73.0 | 94.5 | 98.5 | 94.0 |
| Bottom | 14.0 | 86.0 | 99.5 | 95.0 |
| Top | 13.0 | 62.0 | 100.0 | 96.0 |

Table 10: Ablation experiment results on LLaMA-2-7B showing the performance of pruning different layers. Middle layers yield the best compliance on pseudo-harmful benchmarks, indicating their critical role in distinguishing between harmful and pseudo-harmful instructions.

| Setting | default | layer 14 | layer 15 | layer 16 | layer 17 | layer 18 |
|---------|---------|----------|----------|----------|----------|----------|
| C.R. | 83.5 | 96 | 93 | 91 | 91 | 96 |

Table 11: Performance of Qwen2.5-7B across different middle layers in OR-Bench.

### D.4 ABLATION ANALYSIS OF PRUNING ACROSS DIFFERENT LAYERS

**The superiority of the middle layer.** Ablation experiments (Table 10) conducted on LLaMA-2-7B show that pruning the middle layers yields better performance than pruning the top or bottom

layers. The middle layers, defined as layers [13, 16], significantly outperform both the bottom layers [0, 3] and the top layers [28, 31] on the pseudo-harmful dataset (OR-Bench Hard) and maintain competitive performance on harmful benchmarks (AdvBench and JailbreakBench). Specifically, pruning in the middle layers leads to a marked improvement in compliance rate on OR-Bench Hard and PHTest, suggesting that middle layers play a critical role in distinguishing between harmful and pseudo-harmful instructions. On the other hand, pruning at the bottom or top layers does not show the same level of improvement, with the performance on OR-Bench Hard being considerably lower for both bottom and top pruning.

This finding indicates that middle layers have the optimal capacity for handling safety-related tasks, and pruning them effectively reduces over-refusal without sacrificing overall model performance on genuinely harmful instructions.

**The impact of specific selection of middle layer.** As shown in Table 11, we demonstrate the effect of pruning near the originally set pruning layers. It can be observed that pruning in the middle layers generally contributes to alleviating over-refusal; thus, the selection of pruning layers is not very sensitive.

### D.5 QUANTITATIVE RESULTS FOR DEPLOYMENT PRACTICALITY

Representation editing demands extra storage of fixed intervention vectors, which users must store with the model during loading or distribution and thus raises usage and maintenance costs. By contrast, ProSafePrune generates a fully independent model post-pruning, with no need for additional accompanying files, making migration and deployment more convenient. We further report inference times on OR-Bench-Hard-1K (200 samples, max generation length 256): Self-CD (43 mins), SCAN (20 mins), Surgical (21 mins) and ProSafePrune (only 16 mins). Self-CD incurs extreme overhead due to multiple comparative inferences; feature intervention methods require tensor flow truncation and insertion, which disrupts inference framework optimization and causes extra latency.

