# OpenReview forum: "ProSafePrune: Projected Safety Pruning for Mitigating Over-Refusal in LLMs"
_ICLR.cc/2026/Conference — ICLR 2026 Poster_

### Official Review · Reviewer_jCyL · 2025-10-17

**Soundness:** 3
**Presentation:** 2
**Contribution:** 2
**Rating:** 6
**Confidence:** 2

**Summary:**

This paper proposes ProSafePrune, a low-rank parameter pruning method to mitigate over-refusal in large language models. By identifying and pruning harmful components within pseudo-harmful subspaces, the approach reduces false refusals while preserving genuine safety. Experiments on multiple LLMs show improved compliance and balanced safety performance.

**Strengths:**

1. The proposed method addresses an important and timely issue in LLM alignment with clear motivation and practical relevance.

2. This paper proposes a lightweight, training-free pruning framework that is theoretically grounded and computationally efficient.

**Weaknesses:**

1. The claimed “cognitive bias in internal representations” remains largely conceptual — the paper lacks deeper interpretability analysis or causal evidence supporting this mechanism.

2. The proposed pruning relies on white-box parameter access, limiting applicability to real-world LLMs.

3. The novelty is incremental, as the idea of subspace-based low-rank pruning has been explored ; this paper mainly repurposes it for the over-refusal context.

4. While benchmarks show strong quantitative results, the evaluation misses human evaluation.

**Questions:**

1. The paper attributes over-refusal to “over-harmful encoding” in internal representations. Could the authors elaborate on how this relates to recent work on safe unlearning? Conceptually, is ProSafePrune pruning a similar subspace that unlearning methods aim to erase, or does it operate on a different mechanism?

2. How were the pruning hyperparameters (e.g., λ and rank r) selected, and how sensitive are the results to these choices?

3. Would combining ProSafePrune with post-training alignment or inference-time steering further improve the safety–utility trade-off?

---

> ### Author Response · Authors · 2025-11-20
>
> Thank you for the positive and encouraging feedback. We appreciate the reviewer’s recognition of the importance of the problem, as well as the practical value and theoretical grounding of our lightweight, training-free pruning framework.
>
> **Regarding the interpretability analysis of cognitive bias**
>
> The cognitive bias we describe is a metaphorical notion: through its internal encoding process, the model behaves as if it “interprets” or “perceives” pseudo-harmful instructions as genuinely harmful, resulting in over-harmful representations. This encoding-driven misclassification was already demonstrated and visualized in Section 3.1 of the original paper using probing methods.
>
> To further explain this bias more intuitively, we conducted a word mapping experiment.
>
> We have added a more detailed discussion in Section 5.2. We apply the Logits Lens method to analyze word–representation mappings. This method decodes early model layers to examine the association between intermediate activations and target vocabulary. We compute the total probability of typical refusal-related words—such as “I”, “sorry”, “cannot”—which are frequently used as opening tokens of refusal responses. This provides an intuitive measure of the model’s internal cognitive mapping. As shown in Figure 6 (solid lines), Llama2-7B displays clear probability differences between harmful and harmless datasets in deeper layers. However, for the pseudo-harmful dataset, the model exhibits a pronounced cognitive bias: ideally, such inputs should resemble harmless queries, yet the model assigns unexpectedly high refusal-related probabilities. After applying ProSafePrune, the dangerous-mapping probability for pseudo-harmful data drops significantly, indicating that our method effectively corrects the model’s internal cognitive bias—i.e., the model becomes more inclined to interpret these queries as harmless. This experiment provides direct causal evidence supporting our mechanism. In addition, to strengthen the overall narrative, Section 5.2 now also discusses the connection between alignment tax and general capability degradation, while Section 5.3 analyzes the multidimensional nature of false refusal.
>
> **Regarding the white-box setting**
>
> Nearly all prior work in this research area requires white-box access. Black-box settings only allow adjustments at the prompt level, similar to simple jailbreak attempts. Such methods are generally weak, non-robust, and lack transferability—prompts tuned for one model seldom work reliably on others. In contrast, our method is shown to be applicable across multiple model architectures, demonstrating far better generality.
>
> **Regarding the claim that our novelty is incremental**
>
> The low-rank formulation in our method should be viewed primarily as a mathematical tool, rather than the essence of our contribution. While our method builds on the general mathematical idea of low-rank structure, its problem formulation, motivation, theoretical framing, optimization objective, and behavioral focus are fundamentally different from prior pruning work. Previous pruning research has not addressed false refusal or analyzed misalignment at the level of over-harmful encoding. Our work is, to our knowledge, the first to identify and study the low-rank structure underlying this phenomenon, and to use pruning as a means of over refusal repair. In this sense, although the underlying tool is familiar, the problem setting, methodological intention, and resulting alignment implications are new. We appreciate the reviewer’s thoughtful feedback and hope this clarification helps convey the distinct contribution of ProSafePrune.
>
> **Regarding human evaluation, as suggested**
>
> We included qualitative examples in Appendix C.4 that clearly illustrate the effectiveness of our method.

---

> > ### Author Response · Authors · 2025-11-20
> >
> > **Regarding the relationship to safety unlearning**
> >
> > Existing unlearning methods aim to remove the model’s true harmful knowledge, often by optimizing against datasets containing dangerous content (e.g., violence, hate, privacy violations) so that the model “forgets” how to generate such harmful responses. Their goal is to suppress real harmful capability. In contrast, our notion of over-harmful encoding does not concern harmful knowledge; instead, it refers to the phenomenon where the model mistakenly embeds harmless inputs into harmful-like internal directions, leading to false refusal. Our goal is not “to make the model safer,” but rather “to prevent the model from misclassifying harmless queries as harmful”—a nearly opposite objective. As for whether the subspace removed by ProSafePrune overlaps with the subspace targeted by safety unlearning, we believe the two are largely distinct: our subspace is low-rank and specifically tied to harmfulness encoding, whereas harmful knowledge learned during pretraining is dense, information-rich, and unlikely to be captured by a low-rank structure.
> >
> > **Regarding the choice of λ.**
> >
> > λ exhibits a clear linear behavior, effectively functioning as a controllable switch. As shown in Equation (3) and Figure 11, setting λ = 0 corresponds to no pruning at all, whereas increasing λ leads to an approximately linear increase in the response rate on the pseudo-harmful dataset. This strongly supports our claim that the harmfulness-related components encoded in model parameters possess a linear structural nature.
> > In practice, choosing λ = 0.9 or 1.0 yields a good balance. Smaller values of λ make the model more conservative.
> >
> > **Regarding the selection of rank r.**
> >
> > As discussed in Appendix D.1, we performed pruning-rank experiments on Llama2-7B. When r is too small (e.g., 4 or 8), the model indeed achieves high response rates on the pseudo-harmful dataset, but its refusal rate on the truly harmful dataset drops significantly  (as shown in Figure 10). This phenomenon can be explained by Figure 9: when the rank is very small, the overlap between pseudo-harmful and harmful subspaces becomes large, meaning that after the projection of Eq. (1), a substantial amount of energy remains, causing the pruning to inadvertently remove weights that encode genuine harmful signals.
> > Conversely, overly large ranks also degrade performance. Thus, selecting r = 16 is appropriate—it corresponds to the rank where the harmful–pseudo-harmful subspace overlap reaches its minimum (as shown in Figure 9).
> >
> > **Regarding combining ProSafePrune with other methods**
> >
> > In principle the pruned model can be treated as an independent model and integrated into other safety frameworks. We have tested this by directly replacing the backbone in their released code with our pruned model. Unfortunately, although performance improved compared to the unmodified model, the results were weaker than using ProSafePrune alone. We believe the main reason is hyperparameter mismatch between the pruned model and their fixed settings. With careful retuning, integration should be feasible, as there is no fundamental incompatibility between the approaches.

---

> > > ### Comment · Reviewer_jCyL · 2025-11-22
> > > **Thanks**
> > >
> > > I have read the authors’ response carefully and appreciate the additional clarifications. Although I am not an expert in unlearning, the paper remains solid in its contributions and technical quality. I therefore maintain my positive score.

---

> > > > ### Author Response · Authors · 2025-11-26
> > > > **Thanks**
> > > >
> > > > Thank you for your thoughtful feedback and for taking the time to carefully review our paper. We greatly appreciate your positive evaluation and the constructive comments.

---

### Official Review · Reviewer_bZK9 · 2025-10-29

**Soundness:** 3
**Presentation:** 3
**Contribution:** 2
**Rating:** 4
**Confidence:** 3

**Summary:**

This paper focuses on mitigating over-refusal in LLMs. The authors argue that this behavior comes from harmful encodings overlapping with pseudo-harmful ones in the model’s latent space. To address this, they propose a training-free and subspace-based low-rank pruning method. The key idea is to use (truncated) SVD to identify and remove directions in parameter space that push pseudo-harmful instructions toward harmful representations while not compromising the model’s refusal and general abilities. Experiments across model scales in Llama and Qwen families show lower false rejection rates and stable safety performance, while maintaining or improving general-task performance.

**Strengths:**

1. **Simple and elegant method.**
- Low-rank pruning based on subspace overlap is lightweight and does not require retraining or inference-time overhead.

2. **Strong empirical results.**
- Overall, the proposed method improves compliance on multiple pseudo-harmful benchmarks while maintaining safety on harmful ones (Table 1, Figure 4).

3. **Good theoretical evidence support.**
- The energy-removal guarantee in Theorem 3.2 supports the claim that pruning minimally disrupts the model’s general capacity.

4. **Interesting insights connected to alignment tax.**
- The additional experiments in Section 5.2 add conceptual depth to understanding how overly strong safety alignment can constrain model capabilities.

**Weaknesses:**

1. **Suspiciously large gap on OR-Bench despite training on it.**
- In Table 1, the proposed method shows promising performance on average, but for LLaMA‑3‑8B on OR-Bench the score (71.0) is far lower than Self-CD (86.0). Since the pseudo-harmful subspace is built using OR-Bench data, this is a bit suspicious and raises the question of whether the subspace construction is stable.

2. **The results would be more robust with additional evidence.**
- For example, confidence intervals or std errors for model performances for Table 1 and Figure 4, how pruning rank and λ are selected for Table 3, and potential evaluation on do_sample=True to give a better sense of the method’s stability under more realistic decoding settings.

3. **Small evaluation samples.**
- The model performances are reported on a few hundred examples per dataset (according to Section B.2), which has the concern that small fluctuations could produce the reported gains.

**Questions:**

- Qualitative analysis on cases where the proposed method helps or hurts.
- Submodule-level insight. The pruning is done on Q/K/V/O/MLP modules with no analysis on which parts of the network contribute most to improvements.

---

> ### Author Response · Authors · 2025-11-20
>
> Thank you for the positive and encouraging feedback. We appreciate the reviewer’s recognition of our method’s simplicity, strong empirical results, solid theoretical support, and the insights connecting our findings to alignment tax. Your comments nicely highlight the key contributions of our work.
>
> **Regarding concerns about stability due to the large gap on OR-Bench**
>
> It may be a misunderstanding, and we would like to clarify this.
> The stability of a subspace–based method should primarily be evaluated relative to the default model, because pruning is applied on top of the default model. If pruning resulted in degraded performance relative to the default model, that would indeed indicate instability; however, this does not occur in any of our experiments.
>
> Self-CD, on the other hand, is a baseline, not a reference point for stability.
>
> More importantly, as shown in Table 1, Self-CD exhibits a dramatic drop in refusal rates on harmful datasets (AdvBench: 99 → 87.5; JBB: 95 → 73), significantly worse than our method and all other baselines. In other words, its high compliance rate is achieved by substantially lowering harmful-scenario safety scores, which is undesirable for this task.
> We include this result as an example of over-correction, whereas ProSafePrune achieves a superior balance as reflected by the Avg. T.S. metric.
>
> **Regarding standard errors under do_sample=True as a more realistic decoding setting**
>
> We present this analysis in Table 6 of Appendix B.3.
> We conduct three independent sampling runs (temperature = 1), and report both the mean and variance.
> The results show that our method remains highly stable under realistic decoding, as summarized in the table below:
>
> | or-bench | PHTest | XSTest | OKTest | AdvBench | JBB |
> |----------|--------|--------|--------|----------|-----|
> | 73.8 ± 0.8 | 95 ± 0.6 | 88.4 ± 0.5 | 80.8 ± 1.0 | 98.5 ± 0.1 | 93.7 ± 0.4 |
>
>
> **Regarding the choice of λ.**
>
> λ exhibits a clear linear behavior, effectively functioning as a controllable switch. As shown in Equation (3) and Figure 11, setting λ = 0 corresponds to no pruning at all, whereas increasing λ leads to an approximately linear increase in the response rate on the pseudo-harmful dataset. This strongly supports our claim that the harmfulness-related components encoded in model parameters possess a linear structural nature.
> In practice, choosing λ = 0.9 or 1.0 yields a good balance. Smaller values of λ make the model more conservative.
>
> **Regarding the selection of rank r.**
>
> As discussed in Appendix D.1, we performed pruning-rank experiments on Llama2-7B. When r is too small (e.g., 4 or 8), the model indeed achieves high response rates on the pseudo-harmful dataset, but its refusal rate on the truly harmful dataset drops significantly  (as shown in Figure 10). This phenomenon can be explained by Figure 9: when the rank is very small, the overlap between pseudo-harmful and harmful subspaces becomes large, meaning that after the projection of Eq. (1), a substantial amount of energy remains, causing the pruning to inadvertently remove weights that encode genuine harmful signals.
> Conversely, overly large ranks also degrade performance. Thus, selecting r = 16 is appropriate—it corresponds to the rank where the harmful–pseudo-harmful subspace overlap reaches its minimum (as shown in Figure 9).
>
> **Regarding small sample evaluation concerns**
>
> Given the substantial performance improvements in Table 1, it is unlikely that the gains arise from small-sample statistical noise.
>
> Moreover, one of our baselines—SURGICAL (ICLR 2025)—uses only 128 examples in its original implementation, which the community already considers sufficient for demonstrating improvements in this task.
>
> Nevertheless, we understand the reviewer’s concern about robustness. Therefore, we additionally report model performance under larger evaluation sample sizes to further rule out small-sample variance:
>
> ### Llama2-7B
>
> | Setting | or-bench-hard-1k (1319) | PHTest (2077) |
> |---------|---------------------------|----------------|
> | init    | 12.6                      | 61.9           |
> | prune   | 86                        | 96.4           |
>
>
> ### Qwen2.5-7B
>
> | Setting | or-bench-hard-1k (1319) | PHTest (2077) |
> |---------|---------------------------|----------------|
> | init    | 81.8                      | 96.9           |
> | prune   | 89.5                      | 97.6           |
>
>
> These results confirm that the advantages of ProSafePrune persist across larger evaluation sets.

---

> ### Author Response · Authors · 2025-11-20
>
> **Regarding qualitative case studies**
>
> We provide several qualitative examples in Figure 8 of Appendix C.4, which intuitively illustrate the effectiveness of our method.
>
> **Regarding submodule-level insights**
>
> We have added this analysis in Section 5.3, Table 2, reproduced below:
>
> | default | Q    | K    | V    | O    | MLP  | ALL |
> |---------|------|------|------|------|------|-----|
> | 11      | 10.5 | 11.5 | 30.5 | 16   | 19   | 73  |
>
>
>
>
> The results reveal that pruning the V weight matrix significantly alleviates the model's over-refusal problem, increasing adherence from 11 to 30.5. In contrast, pruning other submodules yields only marginal improvements. Overall, pruning a single submodule is far less effective than pruning the entire layer, suggesting that the over-refusal behavior likely arises from interactions among submodules, rather than the contribution of any single module.

---

> ### Comment · Reviewer_bZK9 · 2025-11-25
>
> Thank you for your comprehensive response. I believe that this paper makes good contribution to LLM safety and overrefusal. The method is elegant and simple, and the results are promising and well supported. I've raised my score to 6.

---

> > ### Author Response · Authors · 2025-11-26
> > **Thanks**
> >
> > Thank you very much for your detailed and thoughtful feedback. We truly appreciate your recognition of the contribution our paper makes to LLM safety and overrefusal. Your updated score and encouragement are greatly appreciated, and we are grateful for your support in enhancing the quality of our work.

---

### Official Review · Reviewer_htFg · 2025-10-31

**Soundness:** 3
**Presentation:** 3
**Contribution:** 2
**Rating:** 6
**Confidence:** 4

**Summary:**

This paper studies the over-refusal issue in LLMs, where harmless instructions are incorrectly rejected due to overlapping safety and harmful feature representations. The authors introduce ProSafePrune, a subspace-projected low-rank pruning framework that removes harmful components from key layers, effectively mitigating over-refusal while preserving genuine safety behavior and slightly improving overall task performance across various models.

**Strengths:**

1. Addressing the over-refusal issue in LLMs through parameter subspace-projected low-rank pruning is a novel approach, and the extensive experiments convincingly demonstrate its effectiveness.

2. The paper is well organized and clearly written, making it easy to follow.

**Weaknesses:**

1. The advantage of the proposed parameter subspace low-rank pruning approach over representation editing in mitigating over-refusal in LLMs is not convincingly demonstrated. Incorporating additional theoretical analysis or interpretability-oriented empirical studies could strengthen the comparative argument and better substantiate the contribution of this work.

2. The finding that the proposed framework does not degrade, and even slightly improves, the general capabilities of LLMs is intriguing. However, the underlying causes of this improvement remain unclear. A more in-depth discussion would help elucidate the underlying mechanisms of this phenomenon.

3. The evaluation is currently limited to LLMs within the 7B–14B parameter range. Assessing the scalability and effectiveness of the proposed framework on larger models (e.g., Qwen3-32B, Llama-3.1-70B) would further validate its robustness and practical applicability.

**Questions:**

1. The finding that the proposed framework does not degrade, and even slightly improves, the general capabilities of LLMs is intriguing. What do the authors consider to be the underlying causes of this improvement?

---

> ### Author Response · Authors · 2025-11-20
>
> Thank you for the positive feedback. We appreciate the reviewer’s recognition of the novelty of our approach and the clarity and organization of the paper. Your comments are encouraging and helpful.
>
> **On the advantages of low-rank pruning over representation editing for mitigating false refusal**
>
> We conducted a demonstration experiment that can to some extent verify the superiority of subspace pruning. We also introduced other practical advantages below. We have added a more detailed discussion in Section 5.3.
>
> Most existing representation-editing methods operate by computing a difference vector between safe and pseudo-harmful features and then removing this direction from intermediate activations. Conceptually, these approaches assume that false refusal can be characterized by a single vector direction.
>
> To test whether this assumption holds, we construct false-refusal directions for multiple common categories of pseudo-harmful data (e.g., sexual, violence, privacy, etc.). We then follow standard single-vector editing procedures and average these category-specific vectors into a global “refusal direction.” If this averaged vector is highly collinear with each category vector, the single-direction assumption would be valid.
>
> However, as shown by our experimental results (Table 4 or table below), the collinearity among category-specific refusal directions is consistently low, indicating that false refusal is not a single-dimensional phenomenon, but instead exhibits clear multi-dimensional structure.
>
> | Category   | sexual | hate | harassment | privacy | illegal | violence | unethical | self-harm | harmful | deception |
> |------------|--------|------|------------|---------|---------|----------|-----------|-----------|---------|-----------|
> | Cosine     | -0.64  | 0.67 | -0.57      | 0.73    | -0.61   | 0.68     | -0.61     | -0.65     | -0.60   | -0.55     |
>
>
> This implies that single-vector editing methods are fundamentally limited in representational capacity. In contrast, ProSafePrune leverages low-rank subspace projection, which is capable of capturing this multi-dimensional structure of erroneous refusal signals, explaining the significant performance improvements observed in Table 1. We also quantify the energy ratio of the projections: projecting difference vectors onto the averaged single vector captures only 56% of the energy, whereas projecting onto the low-rank subspace (r = 16) captures nearly 100%.
>
> In addition, ProSafePrune has practical deployment advantages over representation-editing methods. Editing-based approaches require storing an additional intervention vector; users must load and distribute this vector alongside the model, adding maintenance overhead. In contrast, the pruned model produced by ProSafePrune is fully self-contained, requiring no external files, making it far more convenient to deploy.
>
> We also report inference-time comparisons on OR-Bench-Hard-1K (200 samples, max generation length 256):
>
> Self-CD: 43 min
> SCAN: 20 min
> Surgical: 21 min
> ProSafePrune: 16 min
>
>
> Self-CD incurs very high cost due to repeated forward passes.
> representation-editing methods must truncate and modify tensor flows, which breaks low-level inference optimizations and introduces additional latency.
> In contrast, ProSafePrune preserves the exact same forward path as the original model and therefore incurs no extra inference overhead.
>
> Overall, in terms of theoretical expressiveness, practical performance, and deployment efficiency, ProSafePrune aligns more naturally with the true multi-dimensional structure of false refusal and is better suited for real-world application than existing single-vector editing approaches.
>
> **Regarding evaluation on larger models.**
>
> We have added the corresponding experiments in Appendix C.3 (see the table below).
> ### Llama2-70B
>
> | or-bench-hard-1k | PHTest | XSTest | OKTest | AdvBench | JBB |
> |------------------|--------|--------|--------|----------|-----|
> | 6.5              | 73.5   | 67.2   | 63     | 100      | 90  |
> | 68.5             | 91     | 81.2   | 72.3   | 98.5     | 86  |
>
> ### Qwen3-32B
>
> | or-bench-hard-1k | PHTest | XSTest | OKTest | AdvBench | JBB |
> |------------------|--------|--------|--------|----------|-----|
> | 75.5             | 94     | 96     | 93     | 99.5     | 93  |
> | 80               | 94     | 97.2   | 93.2   | 99.5     | 94  |
>
> The results show that our method scales effectively to models with larger parameter counts. Based on these findings, we believe our approach can be applied to models of arbitrary sizes.

---

> ### Author Response · Authors · 2025-11-20
>
> **On the slight improvement in general capabilities**
>
> We added this analysis and supporting experiments in the last part of Section 5.2.
>
> Qi et al. (Safety Alignment Should Be Made More Than Just a Few Tokens Deep, ICLR25 outstanding paper) point out that current safety alignment largely focuses on “shallow alignment”, meaning that alignment primarily affects the first few output tokens (e.g., forcing refusal prefixes such as “I cannot”). During alignment training, the model may overfit to these fixed refusal words, leading to a collapse where even harmless queries receive abnormally high refusal-word probabilities, thereby suppressing normal expression.
>
> Using the Logits Lens method (decoding early layers), we performed mapping analysis on the fully harmless general-capability dataset CommonQA, as shown in Figure 6 (dashed lines). Comparing Llama2-7B and Qwen-2.5-7B—where the latter has much stronger general capabilities—we observe that Qwen’s final-layer refusal-word probability is nearly zero, while Llama2-7B exhibits a substantial value. After pruning, this refusal-mapping probability in Llama2-7B significantly decreases, accompanied by improvements in general-capability performance.
>
> Simply put, for a model with strong reasoning ability and without over-alignment—such as Qwen—the internal harmfulness probability for completely harmless instructions should be close to zero. Otherwise, from a practical perspective, the model is allocating probability mass that should belong to the correct reasoning chain to meaningless refusal tokens; intuitively, this reflects an overly cautious model that hesitates to provide an answer.
>
> This provides a plausible explanation for why ProSafePrune yields slight improvements in general capabilities. Furthermore, our results suggest that the overfitting to refusal words during alignment training is largely carried by the low-rank structure identified and removed by ProSafePrune.

---

> ### Author Response · Authors · 2025-11-26
>
> Thank you for your previous feedback. The reply phase is coming to an end, and we noticed that we have not yet received your feedback on our clarifications and additional results. If you have any questions or need further clarifications, please feel free to let us know, and we will do our best to address them. We look forward to your response.

---

### Official Review · Reviewer_QSHo · 2025-11-02

**Soundness:** 2
**Presentation:** 3
**Contribution:** 3
**Rating:** 4
**Confidence:** 3

**Summary:**

This paper introduces ProSafePrune, a training-free framework designed to mitigate the "over-refusal" phenomenon in LLMs. The authors identify the root cause as a cognitive bias within the model's internal representation space, where pseudo-harmful instructions are "over-harmfully encoded". ProSafePrune uses subspace projection via truncated SVD to identify and prune low-rank components corresponding to harmful amplification directions in the most discriminative layers.

**Strengths:**

1. This paper shifts the focus from activation-level interventions to directly modifying the model parameters to mitigate the over-refusal phenomenon, which is a new direction.
2. ProSafePrune consistently demonstrates superior performance across diverse LLMs (LLaMA-2/3) and a range of over-refusal benchmarks (OR-Bench, PHTest, XSTest, OKTest).

**Weaknesses:**

1. Sensitive to prune layers and hyperparameter lambda: From table 3 and figure 8, there is a close relationship between model performance and careful tuning of those hyperparameters. It is better to explain how ProSafePrune selects the prune layers since authors only claim high-scoring middle layers as candidates for pruning but without presenting the score threshold for selection.
2. Unclear scalability: The experiments focus on models up to 14B parameters. It’s not clear how well the approach would scale to larger models that may have different internal representations.
3. Lack of pruning time report: Although ProSafePrune employs once static pruning, it is better to report the time cost compared to those training-free baselines.

**Questions:**

See above

---

> ### Author Response · Authors · 2025-11-20
>
> Thank you for highlighting these strengths. We appreciate the reviewer’s acknowledgement of the new direction and the strong performance across models and benchmarks.
>
> **Regarding the choice of pruning layers**
>
> We would like to clarify this point: regarding the selection of pruning layers, it was introduced at the end of Section 3.3 in the original paper. Meanwhile, the number of pruning layers was already discussed in the further explanation in Appendix B.3 of the original paper. We do not use a threshold for the selection.
>
> The choice of intermediate layers is not highly sensitive; we determine it by searching within a local window using a small validation set.
>
> More specifically:
> Within a window of 5–10 layers around the highest TS-score layer (depending on model size), we perform a small search using a lightweight validation set consisting of 50 pseudo-harmful and truly harmful samples. The layer with the highest TS score is selected as the pruning layer.
>
> In practice, pruning layers within the high-scoring middle-layer region generally yields strong results. We provide corresponding experiments in Appendix D.4 (Table 11).
> Conversely, pruning at very early or very late layers is much less effective (as shown in Table 10 of Appendix D.4). Therefore, layer selection within the middle region is not highly sensitive.
>
> Regarding the number of pruning layers, we provide further explanation in Appendix B.3.
> For LLaMA models, pruning only a single layer has almost no effect—the impact on outputs is minimal. Thus, for this model family we prune multiple layers, and empirical results show that pruning more layers leads to higher response rates on pseudo-harmful datasets. However, to avoid overly aggressive pruning, we find 4–8 layers (depending on model size) to be an appropriate choice.
>
> For the Qwen family, pruning only 1–2 layers is sufficient to produce strong improvements. As indicated by the initial pseudo-harmful response rates (Tables 1, 7, and 8), LLaMA models exhibit more severe over-refusal, which explains why more pruning layers are required.
>
> **Regarding the choice of λ.**
>
> λ exhibits a clear linear behavior, effectively functioning as a controllable switch. As shown in Equation (3) and Figure 11, setting λ = 0 corresponds to no pruning at all, whereas increasing λ leads to an approximately linear increase in the response rate on the pseudo-harmful dataset. This strongly supports our claim that the harmfulness-related components encoded in model parameters possess a linear structural nature.
> In practice, choosing λ = 0.9 or 1.0 yields a good balance. Smaller values of λ make the model more conservative.
>
> **Regarding evaluation on larger models.**
>
>  We have added the corresponding experiments in Appendix C.3 (see the table below).
> ### Llama2-70B
>
> | or-bench-hard-1k | PHTest | XSTest | OKTest | AdvBench | JBB |
> |------------------|--------|--------|--------|----------|-----|
> | 6.5              | 73.5   | 67.2   | 63     | 100      | 90  |
> | 68.5             | 91     | 81.2   | 72.3   | 98.5     | 86  |
>
> ### Qwen3-32B
>
> | or-bench-hard-1k | PHTest | XSTest | OKTest | AdvBench | JBB |
> |------------------|--------|--------|--------|----------|-----|
> | 75.5             | 94     | 96     | 93     | 99.5     | 93  |
> | 80               | 94     | 97.2   | 93.2   | 99.5     | 94  |
>
> The results show that our method scales effectively to models with larger parameter counts. Based on these findings, we believe our approach can be applied to models of arbitrary sizes.
>
> **Regarding pruning time reporting**
>
> Thank you for the suggestion. We have added the pruning-time analysis to Section 5.3.
> Pruning itself is extremely fast—on an A40 GPU, pruning and saving a 7B model takes less than 10 seconds.
>
> We also report inference times of all baselines on Llama2-7B evaluated on OR-Bench-Hard-1K (sampled 200 examples, max generation length 256):
>
> Self-CD: 43 min
> SCAN: 20 min
> Surgical: 21 min
> ProSafePrune: 16 min
>
>
> Self-CD requires multiple forward passes, resulting in very high latency.
> representation -intervention-based methods require cutting the tensor flow and injecting intervention vectors during inference; because existing inference frameworks heavily optimize low-level tensor flows, such interruption introduces nontrivial overhead.
>
> In contrast, our method introduces zero additional inference cost. After pruning, the model operates exactly like the original architecture, without any extra runtime operations.

---

> ### Author Response · Authors · 2025-11-26
>
> Thank you for your previous feedback. The reply phase is coming to an end, and we noticed that we have not yet received your feedback on our clarifications and additional results. If you have any questions or need further clarifications, please feel free to let us know, and we will do our best to address them. We look forward to your response.

---

### Author Response · Authors · 2025-11-20

We thank all the reviewers for their recognition of our work!

In summary, through this rebuttal we have clarified key design decisions and provided additional evidence requested by the reviewers. Specifically, we (1) clarified the design choices of our method, including how we select pruning layers, the number of layers, the rank r, and the scaling factor λ, and showed that performance is generally robust within the high-scoring middle-layer region; (2) added experiments on larger models (Llama2-70B and Qwen3-32B), demonstrating that ProSafePrune scales to higher parameter regimes; (3) reported pruning time and inference-time comparisons, showing that our method is essentially cost-free at inference compared to representation-editing and self-CD baselines; (4) provided stability results under stochastic decoding (do_sample=True), and additional evaluations with larger sample sizes to address robustness concerns.

On the analysis side, we (5) introduced a new comparison against single-vector representation editing, including category-wise cosine analysis and projection-energy measurements, showing that false refusal is inherently multi-dimensional and better captured by our low-rank subspace approach; (6) added a Logits Lens–based interpretability study of cognitive bias and alignment tax, and discussed why ProSafePrune can slightly improve general capabilities; (7) included submodule-level pruning analysis and qualitative case studies; and (8) clarified the conceptual distinction between our “over-harmful encoding” mechanism and safety unlearning, as well as the relationship between our white-box setting and other alignment frameworks.

In response to several valuable suggestions raised by the reviewers that were not fully addressed in the original submission, we have added the corresponding analyses and experiments. A revised PDF has been uploaded, with all newly added content highlighted in blue.

---

### Author Response · Authors · 2025-12-01
**Summary**

We would like to express our sincere gratitude to all reviewers for their time and valuable feedback. Overall, all reviewers have recognized that our method is novel, simple and elegant, with sufficient theoretical support and strong experimental validation. We have revised the new version of the manuscript to address the constructive suggestions put forward by the reviewers. **The main body of the paper remains unchanged**, as the reviewers' comments primarily focus on additional analyses—these have been included in the "FURTHER ANALYSIS" section of the paper, with part of the content placed in the appendix. While ablation studies on some hyperparameters were already present in the original manuscript, certain descriptions referencing the appendix were scattered in the theoretical section. To improve structural clarity, we have centralized these contents in the "FURTHER ANALYSIS" section.

Prior to the OpenReview information leak, Reviewers **bZK9** and **jCyL** had responded to our revisions, while Reviewers **QSHo** and **htFg** had not had the opportunity to do so. We wish to clarify that some reviewers had already increased their scores at that stage, with the scores being **4, 6, 6, and 6**. Most reviewers have expressed positive evaluations of our work. We sincerely request that AC take these score increases into consideration.

Below is a detailed explanation of the reviewers' comments and the revisions made to the manuscript:

**For Reviewer QSHo (score: 4)**

This is the **only reviewer** who provided a negative rating and has yet to respond to our clarifications. The suggestions made by this reviewer were few, and we believe there may be some misunderstandings and non-essential suggestion.

The reviewer’s primary concern was the perceived lack of explanation regarding the selection of pruning layers, which may be a **misunderstanding**. In fact, the selection of pruning layers was introduced at the end of Section 3.3 in the original manuscript, and the number of pruning layers was already discussed in the further explanation provided in Appendix B.3 of the original paper. We do not use a threshold for this selection.

Regarding the comment on "unclear scalability," we had already evaluated models across different architectures (Llama 2/3 and Qwen) with parameter sizes of 7B/8B/13B/14B, which constitutes a comprehensive assessment. Most reviewers, including this one, found our experimental results to be strong and extensive. While evaluating larger models could enhance our study, it is not a significant flaw. In fact, **nearly all baseline** evaluations in this field focus primarily on this parameter range, and this is **already sufficient** to demonstrate the effectiveness of our approach. Nevertheless, we acknowledge the reviewer’s valuable suggestion and have supplemented the results of Qwen3-32B and Llama2-70B in the revised manuscript.

Concerning the request for reporting pruning time: We deemed it **unnecessary** because, as the reviewer noted, this is a one-time static operation that takes less than ten seconds in total. However, we have added a discussion on this aspect in the paper, focusing on the comparison of inference times across different methods.

While we have addressed the reviewer’s concerns, unfortunately, we have not received a response from them to date.

**For Reviewer htFg (score: 6)**

The reviewer believes that our work is a novel approach, and the extensive experiments convincingly demonstrate its effectiveness.

Regarding the analysis of the theoretical or interpretability advantages of our method compared to representation editing: Although we have demonstrated experimentally driven advantages through extensive evaluations, we have further added in-depth analyses in the "FURTHER ANALYSIS" section.

The reviewer found it compelling that our method can slightly improve general capabilities and requested additional clarification on this point. We have supplemented relevant interpretive explanations accordingly.

---

> ### Author Response · Authors · 2025-12-01
>
> **For Reviewer bZK9 (revised score: 6)**
>
> The reviewer **highly praised** the advantages (simple and elegant method, strong empirical results, good theoretical evidence support and interesting insights connected to alignment tax) of our method but raised some concerns stemming from potential misunderstandings, primarily regarding stability and evaluation robustness. We have clarified these points to the reviewer, who subsequently **increased their score** (prior to the OpenReview information leak). Additionally, we have incorporated the reviewer’s other valuable suggestions into the manuscript, including submodule-level insights, real-world case demonstrations, and temperature sampling evaluations.
>
> **For Reviewer jCyL (score: 6)**
>
> The reviewer believes that our work addresses an important and timely issue in LLM alignment with clear motivation and practical relevance.
>
> In response to the reviewer’s suggestion, we have added interpretability experiments on word mapping to more intuitively illustrate the "cognitive bias in internal representations" we claimed. We have also addressed and clarified other questions raised by the reviewer. Finally, the reviewer confirmed that **the paper is solid in terms of contributions and technical quality** and maintained their positive rating.

---

### Meta-Review · Area_Chair_mBhq · 2026-01-06

**Summary:**

All four reviewers agree that the paper tackles an important and timely problem: over-refusal in aligned LLMs, where harmless but superficially risky prompts are incorrectly rejected. They view ProSafePrune--a training-free, subspace-projected low-rank pruning method--as simple, elegant, and practically appealing. The method uses pseudo-harmful vs. harmful data to identify a low-rank subspace associated with “over-harmful encoding” in a few critical layers, then prunes directions in that subspace to reduce false refusals while preserving genuine safety and even slightly improving general capabilities.

The main concerns raised were:

- **Hyperparameter and layer selection / stability** (QSHo, bZK9, jCyL): How sensitive is performance to the choice of pruning layers, rank $r$, and scaling factor $\lambda$? Is the method robust, or does it require careful tuning?

- **Scalability to larger models** (QSHo, htFg): Initial experiments were limited to models up to ~14B parameters; reviewers questioned whether the approach scales to larger LLMs.

- **Comparative advantages over representation editing / unlearning** (htFg, jCyL): Reviewers asked for a clearer argument and evidence that low-rank parameter pruning is preferable to single-vector representation editing or safety unlearning approaches.

- **Robustness and evaluation depth** (bZK9): Concerns about small evaluation sets, lack of confidence intervals, stability under stochastic decoding, and an initially puzzling gap vs. Self-CD on OR-Bench, given that OR-Bench is used to build the subspace.

- **Conceptual framing and interpretability** (jCyL): The notion of “cognitive bias in internal representations” needed stronger interpretability/causal support, and the relationship to safety unlearning and white-box assumptions should be clarified.

- **Efficiency reporting** (QSHo, htFg, bZK9): Some reviewers requested pruning time and inference-time comparisons to other methods.

Despite these concerns, three reviewers (htFg, bZK9 after update, and jCyL) scored the paper 6 (marginally above threshold), and QSHo gave a 4 (marginally below but would not mind acceptance). The consensus is that the method is novel in its use of subspace-projected pruning for over-refusal, well-motivated, and empirically strong.

**Reviewer Concerns:**

The authors’ rebuttal and revised submission directly address most of the substantive concerns:

- **Hyperparameter & layer selection / stability.** The authors provide detailed procedures for selecting pruning layers (searching within a window around the highest TS-score layer using a small validation set) and discuss why performance is generally robust within high-scoring middle layers. They add ablations on pruning rank $r$ showing that $r=16$ corresponds to minimal harmful–pseudo-harmful subspace overlap, and they show that $\lambda$ behaves like a nearly linear “strength knob.” Additional results under do_sample=True with multiple runs demonstrate stability under stochastic decoding.

- **Scalability.** New experiments on Llama2-70B and Qwen3-32B show that ProSafePrune continues to reduce false refusals and preserve safety at larger scales, alleviating concerns about scalability.

- **Comparisons vs. representation editing and unlearning.** The rebuttal adds analyses comparing ProSafePrune with single-vector editing: cosine-similarity and projection-energy experiments show that false-refusal signals are multi-dimensional and poorly captured by a single vector, whereas the low-rank subspace captures nearly all the energy. The authors also argue that their low-rank “over-harmful encoding” subspace is distinct from dense harmful-knowledge subspaces targeted by unlearning, clarifying conceptual differences. They further emphasize deployment advantages (no inference-time overhead, self-contained pruned model).

- **Robustness & evaluation depth.** Larger evaluation sets on OR-Bench-Hard-1k and PHTest confirm that gains persist beyond the smaller samples; confidence intervals/variances under sampling are reported; and qualitative case studies plus submodule-level analyses (showing, e.g., the importance of V matrices) deepen the empirical picture. The OR-Bench gap vs. Self-CD is contextualized by showing that Self-CD severely degrades safety on harmful benchmarks, whereas ProSafePrune maintains high safety, so the lower OR-Bench score is not indicative of instability.

- **Interpretability & “cognitive bias.”** The authors add Logits Lens–based analyses showing that refusal-word probabilities for pseudo-harmful prompts are anomalously high in deeper layers and that ProSafePrune reduces this mapping, offering concrete evidence for the “over-harmful encoding” mechanism and explaining why general capabilities can slightly improve after pruning.

- **Efficiency.** Pruning time (under 10 seconds for a 7B model) and inference-time comparisons vs. Self-CD, SCAN, and Surgical are reported, demonstrating that ProSafePrune effectively adds zero runtime overhead and is faster in practice than representation-editing baselines.

Overall, the rebuttal successfully addresses the major points raised by htFg, bZK9, and jCyL, and does so with substantial new experimental and interpretive evidence.

**Remaining concerns are relatively minor**

The method still assumes white-box access and primarily targets open-source models; while this is standard in the alignment-tuning literature, it limits direct applicability to closed models. Human evaluation remains limited to qualitative case studies rather than large-scale annotation. These limitations are reasonable for a first paper on this technique and do not undermine the main empirical and methodological contributions.

**Reviewer Scores:**

In summary, if rebuttal and discussion continue, three reviewers are clearly positive (6, 6, 6), and the remaining reviewer is at worst neutral-to-borderline but likely to soften or slightly improve their score given the additional evidence. The method is simple, novel in its application of low-rank subspace pruning to over-refusal, empirically strong across multiple model families and sizes, and practically attractive due to its training-free, zero-overhead nature. I therefore recommend acceptance.

---

### Decision · Program_Chairs · 2026-01-26

Accept (Poster)